# POST-TRAINING DETECTION OF BACKDOOR ATTACKS FOR TWO-CLASS AND MULTI-ATTACK SCENARIOS

**Zhen Xiang, David J. Miller & George Kesidis**
School of EECS
Pennsylvania State University
{`zux49, djm25, gik2`}@psu.edu

## ABSTRACT

Backdoor attacks (BAs) are an emerging threat to deep neural network classifiers. A victim classifier will predict to an attacker-desired target class whenever a test sample is embedded with the same backdoor pattern (BP) that was used to poison the classifier's training set. Detecting whether a classifier is backdoor attacked is not easy in practice, especially when the defender is, e.g., a downstream user without access to the classifier's training set. This challenge is addressed here by a reverse-engineering defense (RED), which has been shown to yield state-of-the-art performance in several domains. However, existing REDs are not applicable when there are only *two classes* or when *multiple attacks* are present. These scenarios are first studied in the current paper, under the practical constraints that the defender neither has access to the classifier's training set nor to supervision from clean reference classifiers trained for the same domain. We propose a detection framework based on BP reverse-engineering and a novel *expected transferability* (ET) statistic. We show that our ET statistic is effective *using the same detection threshold*, irrespective of the classification domain, the attack configuration, and the BP reverse-engineering algorithm that is used. The excellent performance of our method is demonstrated on six benchmark datasets. Notably, our detection framework is also applicable to multi-class scenarios with multiple attacks. Code is available at https://github.com/zhenxianglance/2ClassBADetection.

## 1 INTRODUCTION

Despite the success of deep neural network (DNN) classifiers in many research areas, their vulnerabilities have been recently exposed Xu et al. (2020); Miller et al. (2020). One emerging threat to DNN classifiers is a backdoor attack (BA) Li et al. (2020). Here, a classifier will predict to the attacker's target class when a test sample is embedded with the same backdoor pattern (BP) that was used to poison the classifier's training set. On the other hand, the classifier's clean test set accuracy (i.e., on samples without an embedded BP) is largely uncompromised Gu et al. (2019); Chen et al. (2017); Liu et al. (2018b), which makes the attack difficult to detect.

Early BA defenses aim to cleanse the possibly poisoned training set of the victim classifier Tran et al. (2018); Chen et al. (2018). But deployment of these defenses is not feasible when the defender is the user/consumer of the classifier, without access to its training set or to any prior knowledge of the backdoor pattern Wang et al. (2020). A major defense approach suitable for this scenario is a reverse-engineering defense (RED). In general, a RED treats each class as a putative BA target class and trial-reverse-engineers a BP. Then, detection statistics are derived from the estimated pattern for each putative target class. If there is a BA, the pattern estimated for the true BA target class will likely be correlated with the true BP used by the attacker, such that the associated statistics will likely be anomalous referenced against the statistics for the other (non-BA) classes Wang et al. (2019); Xiang et al. (2020). Notably, RED-based anomaly detection does not require supervision from clean classifiers trained for the same domain like, e.g., Kolouri et al. (2020).

Although existing REDs have achieved leading performance in many practical detection tasks, a fundamental limitation still remains. RED-based anomaly detection typically requires estimating a null distribution used to assess anomalies, with the number of detection statistics (samples) used to estimate this null a function of the number of classes $K$. Wang et al. (2019) uses $O(K)$ statistics, and Xiang et al. (2020) uses even more statistics – $O(K^2)$. These methods are not applicable for domains with $K = 2$, e.g. sentiment classification Gao et al. (2019b), disease diagnosis Li et al. (2014), etc., because there are insufficient statistics for estimating the null. More generally, their accuracy is affected when the number of classes is small ($K > 2$, but small).

In this paper, we focus on BA detection for the *two-class* (and possibly *multi-attack*) scenario, under two practical constraints: a) the defender has *no access* to the training set of the victim classifier (or to any BP used by the attacker); b) supervision from clean classifiers trained for the same domain is *not available*. This scenario is clearly more challenging than the one considered by existing REDs, for which there is at most one BA target class and a sufficient number of non-target classes to learn an accurate null. As the first to address this difficult problem, our main contributions are as follows:

- We propose a detection framework that involves BP reverse-engineering in order to address constraint a) above. However, instead of performing anomaly detection as existing REDs do, we process each class independently using a novel detection statistic dubbed *expected transferability* (ET). This allows our method to be applicable to the two-class, multi-attack scenario (as well as to the multi-class and (possibly) multi-attack scenario).
- We show that for ET there is a large range of effective choices for the detection threshold, which *commonly* contains a particular threshold value effective for detecting BAs irrespective of the classification domain or particulars of BA. This common threshold is mathematically derived and is based on properties for general classification tasks that have been verified empirically by many existing works; thus, constraint b) that no domain-specific supervision is needed is well-addressed.
- Our ET statistic is obtained by BP reverse-engineering and does not strongly depend on the type of BP, or the particular RED objective function and optimization technique used to reverse-engineer putative BPs. Thus, our detection framework can incorporate existing reverse-engineering techniques and potentially their future advances.
- We show the effectiveness of our detection framework on six popular benchmark image datasets.

## 2 RELATED WORK

**Backdoor Attack (BA).** For a classification task with sample space $\mathcal{X}$ and label space $\mathcal{C}$, a BA aims to have the victim classifier $f : \mathcal{X} \to \mathcal{C}$ predict to the target class $t \in \mathcal{C}$ whenever a test sample $\mathbf{x} \in \mathcal{X}$ is embedded with the backdoor pattern (BP) Gu et al. (2019). BAs were initially proposed for image classification, but have also been proposed for other domains and tasks Xiang et al. (2021a); Zhai et al. (2021); Li et al. (2022). While we focus on images experimentally, our detector is also applicable to other domains. Major image BPs include: 1) an additive perturbation $\mathbf{v}$ embedded by

$$\tilde{\mathbf{x}} = [\mathbf{x} + \mathbf{v}]_c \tag{1}$$

where $||\mathbf{v}||_2$ is small (for imperceptibility) and $[\cdot]_c$ is a clipping function Chen et al. (2017); Zhong et al. (2020); Xiang et al. (2020); or 2) a patch $\mathbf{u}$ inserted using an image-wide binary mask $\mathbf{m}$ via

$$\tilde{\mathbf{x}} = (1 - \mathbf{m}) \odot \mathbf{x} + \mathbf{m} \odot \mathbf{u}, \tag{2}$$

where $||\mathbf{m}||_1$ is small for imperceptibility) and $\odot$ represents element-wise multiplication Gu et al. (2019); Wang et al. (2019); Xiang et al. (2021c). Typically[1], BAs are launched by inserting into the training set a small set of samples labeled to the target class and embedded with the same BP that will be used during testing. Such data poisoning can be achieved *e.g.* when DNN training is outsourced to parties that are possibly malicious Gu et al. (2019). BAs are also stealthy since they do not degrade the classifier's accuracy on clean test samples; hence they are not detectable based on validation set accuracy.

**Backdoor Defense.** Some BA defenses aim to separate backdoor training samples from clean ones during the training process Tran et al. (2018); Chen et al. (2018); Du et al. (2020); Xiang et al. (2019); Huang et al. (2022). However, their deployment is not feasible in many practical scenarios where the defender is a downstream user who has no access to the training process. A family of pruning-based method "removes" the backdoor mapping by inspecting neuron activations (and removing some neurons) Liu et al. (2018a); Li et al. (2021). These methods cause non-negligible degradation to classification accuracy, they may remove neurons even when there is no backdoor, and moreover they do not make explicit detection inferences. Kolouri et al. (2020) and Xu et al. (2021) train a binary classifier to classify models as "with" and "without" BAs; but such training requires clean classifiers from the same domain or a significant number of labeled samples and heavy computation to train these clean classifiers. Chou et al. (2018) and Gao et al. (2019a) can detect triggering of a backdoor by an observed test sample. Unlike these methods, the existing RED-based detector (introduced next) reliably detects backdoors without the need to observe any test samples.

**Reverse-Engineering Defense (RED).** RED is a family of BA defenses that do not need access to the classifier's training set, nor to any prior knowledge of the BP that may be used in an attack.

---

[1]A clean-label BA with a different strategy Turner et al. (2019) is also detectable by our method (Apdx. E).

Without knowing whether the classifier has been backdoor-attacked, a RED trial-reverse-engineers a BP for each putative BA target class (or possibly for each (source, target) BA class pair) using a small, clean dataset independently collected by the defender. Then, for each class, a detection statistic is derived from the estimated pattern and used to infer if the class is a BA target class or not. For example, to detect BAs with an additive perturbation BP (Eq. (1)), Xiang et al. (2020) finds, for each putative target class, the perturbation with the minimum $l_2$ norm that induces a large misclassification fraction to this class when added to images from another class (the source class). Since BPs embedded by Eq. (1) usually have a very small $l_2$ norm for imperceptibility, the pattern estimated for the true BA target class (if a BA exists) will likely have a much smaller $l_2$ norm than for non-target classes. Thus, unsupervised anomaly detection based on these perturbation sizes is performed – when there is one BA target class and a sufficient number of non-target classes, the statistic for the BA target class will likely be detected as an anomaly compared with the others.

Except the RED above proposed by Xiang et al. (2020), existing REDs also include Neural Cleanse (NC) proposed by Wang et al. (2019), which reverse-engineers BPs embedded by Eq. (2) and uses the $l_1$ norm of the estimated mask as the detection statistic. Guo et al. (2019) adds constraints to NC's BP reverse-engineering problem by considering various properties of BPs. Liu et al. (2019) proposed a novel objective function for BP reverse-engineering leveraging the abnormal internal neuron activations caused by the backdoor mapping. Chen et al. (2019) constructs clean images for detection using model inversion. Dong et al. (2021) queries the classifier for BP reverse-engineering to address the "black-box" scenario. Wang et al. (2020) proposes a detection statistic based on the similarity between universal and sample-wise BP estimation.

**Limitation of Existing REDs** The anomaly detection of REDs heavily relies on the assumption that there is a relatively large number of non-target classes, thus providing a sufficient number of statistics to inform estimation of a null distribution. But this assumption does not hold for domains with only two classes. For example, Xiang et al. (2020) and Wang et al. (2019) exploit $O(K^2)$ and $O(K)$ statistics in estimating a null respectively, where $K$ is the number of classes. Both of these methods are unsuitable for 2-class problems ($K = 2$).

## 3 METHOD

Our main goals are detecting whether a given classifier is backdoor attacked or not and, if so, finding out all the BA target classes. We focus on a practical "post-training" scenario: **S1**) The defender has no access to the classifier's training set, nor any prior knowledge of the true BP used by an attacker. **S2**) There are no clean classifiers trained for the same domain for reference; and the defender is not capable of training such clean classifiers. **S3**) The classification domain has two classes that can both be BA targets. While we focus on two-class scenarios in this section, our method is more generally applicable to multi-class scenarios, with arbitrary number of attacks (see experiments in Sec. 4.4).

To address **S1**, our detection framework involves BP reverse-engineering using a small, clean dataset independently collected by the defender, like existing REDs. To address **S3**, unlike existing REDs that perform anomaly detection involving statistics from *all* classes, we inspect each class *independently* using a novel detection statistic called *expected transferability* (ET), which can be empirically estimated for each class independently. To address **S2**, we show that ET possesses a theoretically-grounded detection threshold value for distinguishing BA target classes from non-target classes, *one which depends neither on the domain nor on the attack configuration*. This is very different from existing REDs, for which a suitable detection threshold for their proposed statistics may be both domain and attack-dependent. For example, the range of the $l_1$ norm of the estimated mask used by Wang et al. (2019) depends on the image size. The practical import here is that the detection threshold is a *hyperparameter*, but setting this threshold in a supervised fashion (e.g. to achieve a specified false positive rate on a group of clean classifiers) is generally infeasible due to **S2**. *Use of ET thus obviates the need for such hyperparameter setting.*

In the following, we define ET in Sec. 3.1, and then, the constant threshold on ET for BA detection is derived in Sec. 3.2. Finally, our detection procedure, which consists of ET estimation and an inference step, is in Sec. 3.3. Note that our detection framework is effective for various types of BPs (as will be shown experimentally). Solely for clarity, in this section, we focus on BAs with *additive perturbation* BPs (Eq. (1)). Similar derivation for BPs embedded by Eq. (2) is deferred to Apdx. C.

### 3.1 EXPECTED TRANSFERABILITY (ET)

Consider a classifier $f : \mathcal{X} \to \mathcal{C}$ to be inspected with category space $\mathcal{C} = \{0, 1\}$ and *continuous* sample distribution $P_i$ on $\mathcal{X}$ for class $i \in \mathcal{C}$. For any $\mathbf{x} \in \mathcal{X}$ from any class, the optimal solution to

$$\underset{\mathbf{v}}{\text{minimize}} \, ||\mathbf{v}||_2 \qquad \text{subject to } f(\mathbf{x} + \mathbf{v}) \neq f(\mathbf{x}) \tag{3}$$

is defined as $\mathbf{v}^*(\mathbf{x})$. In practice, (3) can be viewed as a typical BP reverse-engineering problem and solved using methods in existing REDs like Xiang et al. (2020). It can also be practically solved by creating an adversarial example for $\mathbf{x}$ using methods in, e.g., M.-Dezfooli et al. (2016); Carlini & Wagner (2017). We present the following definition for the set of *practical* solutions to (3).

**Definition 3.1. ($\epsilon$-solution set)** For any sample $\mathbf{x}$ from any class, regardless of the method being used, the $\epsilon$-solution set to problem (3), is defined by

$$\mathcal{V}_\epsilon(\mathbf{x}) \triangleq \{\mathbf{v} \in \mathcal{X} \big| ||\mathbf{v}||_2 - ||\mathbf{v}^*(\mathbf{x})||_2 \leq \epsilon, f(\mathbf{x} + \mathbf{v}) \neq f(\mathbf{x})\}, \tag{4}$$

where $\epsilon > 0$ is the "quality gap" of practical solutions, which is usually small for existing methods.

A practical solution to (3) for sample $\mathbf{x}$ may or may not cause a misclassification when embedded in another sample $\mathbf{y}$ from the same class. In the following, we first present the definition regarding such a "transferability" property. Then, we define the ET statistic for any class $i \in \mathcal{C}$.

**Definition 3.2. (Transferable set)** The transferable set for any sample $\mathbf{x}$ and $\epsilon > 0$ is defined by

$$\mathcal{T}_\epsilon(\mathbf{x}) \triangleq \{\mathbf{y} \in \mathcal{X} \big| f(\mathbf{y}) = f(\mathbf{x}), \exists \mathbf{v} \in \mathcal{V}_\epsilon(\mathbf{x}) \, \text{s.t.} \, f(\mathbf{y} + \mathbf{v}) \neq f(\mathbf{y})\}. \tag{5}$$

**Definition 3.3. (ET statistic)** For any class $i \in \mathcal{C} = \{0, 1\}$ and $\epsilon > 0$, considering i.i.d. random samples $\mathbf{X}, \mathbf{Y} \sim P_i$, the ET statistic for class $i$ is defined by $\text{ET}_{i,\epsilon} \triangleq \mathbb{E}\big[\text{P}(\mathbf{Y} \in \mathcal{T}_\epsilon(\mathbf{X})|\mathbf{X})\big]$.

### 3.2 Using ET for BA Detection

Here, we show that for any $i \in \mathcal{C} = \{0, 1\}$ and small $\epsilon$, we will likely have $\text{ET}_{i,\epsilon} > \frac{1}{2}$ when class $(1 - i)$ is a BA target class; and $\text{ET}_{i,\epsilon} \leq \frac{1}{2}$ otherwise. Note that this *constant* threshold does not rely on any specific data domain, classifier architecture, or attack configuration; even for BPs embedded by Eq. (2), the *same* threshold can be obtained following a similar derivation (see Apdx. C).

We first present the following theorem showing the connection between ET and the threshold $\frac{1}{2}$ regardless of the presence of any BA. Then, we discuss the attack and non-attack cases, respectively.

**Theorem 3.1.** For any class $i \in \mathcal{C}$ and $\epsilon > 0$:

$$\text{ET}_{i,\epsilon} = \frac{1}{2} + \frac{1}{2}(\text{P}_{\text{MT},i} - \text{P}_{\text{NT},i}), \tag{6}$$

with $\text{P}_{\text{MT},i}$ a "mutual-transfer probability" and $\text{P}_{\text{NT},i}$ a "non-transfer probability", defined by

$$\text{P}_{\text{MT},i} \triangleq \text{P}(\mathbf{Y} \in \mathcal{T}_\epsilon(\mathbf{X}), \mathbf{X} \in \mathcal{T}_\epsilon(\mathbf{Y})) \quad \text{and} \quad \text{P}_{\text{NT},i} \triangleq \text{P}(\mathbf{Y} \notin \mathcal{T}_\epsilon(\mathbf{X}), \mathbf{X} \notin \mathcal{T}_\epsilon(\mathbf{Y})) \tag{7}$$

respectively, where $\mathbf{X}$ and $\mathbf{Y}$ are i.i.d. random samples following distribution $P_i$ for class $i$.

**1) Non-attack case:** class $(1 - i)$ is *not* a BA target class. We first focus on $\text{P}_{\text{NT},i}$.

**Property 3.1.** For general two-class domains in practice and small $\epsilon$, if class $(1 - i)$ is *not* a BA target class, $\text{P}_{\text{NT},i}$ for class $i$ will likely be larger than $\frac{1}{2}$ (see Sec. 4.3 for empirical support).

For any $i \in \mathcal{C}$, $\text{P}_{\text{NT},i}$ is the probability that two independent samples from class $i$ are mutually "not transferable". For such a pair of samples, the event associated with $\text{P}_{\text{NT},i}$ is that the pattern estimated for one (by solving (3)) does not induce the other to be misclassified and vice versa. Accordingly, if we solve a problem similar to (3) but requiring a common $\mathbf{v}$ that induces both samples to be misclassified, the solution should have a larger norm than the solution to (3) for each of them. Property 3.1 has been verified by many existing works for general classification tasks commonly using highly non-linear classifiers. For example, the universal adversarial perturbation studied by Moosavi-Dezfooli et al. (2017) can be viewed as the solution to problem (3) (in absence of BA) for a group of samples instead of one. Compared with the minimum sample-wise perturbation required for each individual to be misclassified, the minimum universal perturbation required for high group misclassification typically has a much larger norm. Similar empirical results have also been shown by Xiang et al. (2020) and inspired their proposed RED. Despite the evidence in existing works, we also verify this property experimentally in Sec 4.3.

Next, we focus on $\text{P}_{\text{MT},i}$. Note that $\text{P}_{\text{MT},i}$ is upper bounded by $1 - \text{P}_{\text{NT},i}$; thus it will likely be smaller than $\frac{1}{2}$ (and even possibly close to 0) for the non-attack case based on Property 3.1. The asymptotic behavior of $\text{P}_{\text{MT},i}$ for small $\epsilon$ can also be approached by the following theorem. Note that in practice, a small $\epsilon$ is not difficult to achieve, since a solution to problem (3) using, e.g., algorithms for generating adversarial samples Szegedy et al. (2014) usually have a small norm.

---

**Algorithm 1** BA detection using ET statistics.

---

1: **Input**: Classifier $f : \mathcal{X} \to \mathcal{C}$ for inspection; clean dataset $\mathcal{D}_i = \{\mathbf{x}_1^{(i)}, \cdots, \mathbf{x}_{N_i}^{(i)}\}$ for each $i \in \mathcal{C}$.
2: **Initialization**: attacked $= False$; BA_targets $= \emptyset$.
3: **for** each putative target class $t \in \mathcal{C} = \{0, 1\}$ **do**
4:      **Step 1:** Obtain empirical estimation $\widehat{\mathrm{ET}}_{i,\epsilon}$ using $\mathcal{D}_i$ for $i = 1 - t$.
5:      **for** $n = 1 : N_i$ **do**
6:          $\widehat{\mathcal{T}}_\epsilon(\mathbf{x}_n^{(i)}) = \emptyset$; converge $= False$
7:          **while** not converge **do**
8:              Obtain an empirical solution $\hat{\mathbf{v}}(\mathbf{x}_n^{(i)})$ to problem (3) using random initialization.
9:              $\widehat{\mathcal{T}}_\epsilon(\mathbf{x}_n^{(i)}) \leftarrow \widehat{\mathcal{T}}_\epsilon(\mathbf{x}_n^{(i)}) \cup \{\mathbf{x}_m^{(i)} | m \in \{1, \cdots, N_i\} \setminus n, f(\mathbf{x}_m^{(i)} + \hat{\mathbf{v}}(\mathbf{x}_n^{(i)})) \neq f(\mathbf{x}_m^{(i)})\}$
10:             **if** $\widehat{\mathcal{T}}_\epsilon(\mathbf{x}_n^{(i)})$ unchanged for $\tau$ iterations **then**
11:                 converge $\leftarrow True$
12:                 $p_n^{(i)} = |\widehat{\mathcal{T}}_\epsilon(\mathbf{x}_n^{(i)})| / (N_i - 1)$
13:      $\widehat{\mathrm{ET}}_{i,\epsilon} = \frac{1}{N_i} \sum_{n=1}^{N_i} p_n^{(i)}$
14:      **Step 2:** Determine if class $t$ is a BA target class or not.
15:      **if** $\widehat{\mathrm{ET}}_{i,\epsilon} > \frac{1}{2}$ **then**
16:          attacked $= True$; BA_targets $\leftarrow$ BA_targets $\cup \{t\}$
17: **Output**: attacked; BA_targets

---

**Theorem 3.2.** For any class $i \in \mathcal{C}$ with continuous sample distribution $P_i$, $\mathrm{P}_{\mathrm{MT},i} \to 0$ as $\epsilon \to 0$.

In summary, for the non-attack case with small $\epsilon$, we will likely have $\mathrm{P}_{\mathrm{NT},i} \geq \frac{1}{2} \geq \mathrm{P}_{\mathrm{MT},i}$; and thus, $\mathrm{ET}_{i,\epsilon} \leq \frac{1}{2}$ (based on Thm. 3.1) – this will be shown by our experiments. Moreover, in Apdx. B, for a simplified (yet still relatively general) domain and a linear prototype classifier, we show that $\frac{1}{2}$ is a *strict* upper bound of the ET statistic when there is no BA.

**2) Attack case:** class $(1 - i)$ is the target class of a successful BA. Suppose $\mathbf{v}_0$ is the BP for this attack, such that for any $\mathbf{X} \sim P_i$, $f(\mathbf{X}) = i$, while $f(\mathbf{X} + \mathbf{v}_0) \neq f(\mathbf{X})$. Intuitively, $\mathrm{P}_{\mathrm{MT},i}$ will be large and possibly close to 1 because, *different from the non-attack case*, there is a special pattern – the BP $\mathbf{v}_0$ – that could likely be an element of $\mathcal{V}_\epsilon(\mathbf{X})$ and $\mathcal{V}_\epsilon(\mathbf{Y})$ simultaneously, for $\mathbf{X}$ and $\mathbf{Y}$ i.i.d. following $P_i$. In this case, $\mathbf{Y} \in \mathcal{T}_\epsilon(\mathbf{X})$ and $\mathbf{X} \in \mathcal{T}_\epsilon(\mathbf{Y})$ jointly hold (i.e. mutually transferable) by Definition 3.2. Accordingly, $\mathrm{P}_{\mathrm{NT},i}$ which is upper bounded by $1 - \mathrm{P}_{\mathrm{MT},i}$ will likely be small and possibly close to 0; then, we will have $\mathrm{P}_{\mathrm{MT},i} > \mathrm{P}_{\mathrm{NT},i}$ and consequently, $\mathrm{ET}_{i,\epsilon} > \frac{1}{2}$ by Thm. 3.1.

Beyond the intuitive analysis above, the following theorem gives a guaranteed large ET statistic (being exactly 1) in the attack case when the backdoor pattern has a (sufficiently) small norm (which is in fact desired in order to have imperceptibility of the attack).

**Theorem 3.3.** If class $(1-i)$ is the target class of a successful BA with BP $\mathbf{v}_0$ such that $f(\mathbf{X}+\mathbf{v}_0) \neq f(\mathbf{X})$ for all $\mathbf{X} \sim P_i$, and if $||\mathbf{v}_0||_2 \leq \epsilon$, we will have $P(\mathcal{V}_\epsilon(\mathbf{X}) \cap \mathcal{V}_\epsilon(\mathbf{Y}) \neq \emptyset) = 1$ for $\mathbf{X}$ and $\mathbf{Y}$ i.i.d. following $P_i$; and furthermore, $\mathrm{ET}_{i,\epsilon} = 1$.

## 3.3 Detection Procedure

Our detection procedure is summarized in Alg. 1. Basically, for each putative target class $t \in \mathcal{C} = \{0, 1\}$, we estimate the ET statistic $\widehat{\mathrm{ET}}_{i,\epsilon}$ (with a "hat" representing empirical estimation) for class $i = 1-t$, and claim a detection if $\widehat{\mathrm{ET}}_{i,\epsilon} > \frac{1}{2}$. In particular, the core to estimating $\widehat{\mathrm{ET}}_{i,\epsilon}$ is to estimate $\mathrm{P}(\mathbf{Y} \in \mathcal{T}_\epsilon(\mathbf{X}) | \mathbf{X} = \mathbf{x}_n^{(i)})$ for each clean sample $\mathbf{x}_n^{(i)} \in \mathcal{D}_i$ used for detection (line 6-12, Alg. 1). To do so, we propose to find, for each $\mathbf{x}_n^{(i)} \in \mathcal{D}_i$, the subset $\widehat{\mathcal{T}}_\epsilon(\mathbf{x}_n^{(i)})$ which contains *all* samples in $\mathcal{D}_i \setminus \mathbf{x}_n^{(i)}$ belonging to the *transferable set* $\mathcal{T}_\epsilon(\mathbf{x}_n^{(i)})$ of sample $\mathbf{x}_n^{(i)}$. Then, $\mathrm{P}(\mathbf{Y} \in \mathcal{T}_\epsilon(\mathbf{X}) | \mathbf{X} = \mathbf{x}_n^{(i)})$ can be estimated by $p_n^{(i)} = |\widehat{\mathcal{T}}_\epsilon(\mathbf{x}_n^{(i)})| / (|\mathcal{D}_i| - 1)$ (line 12, Alg. 1). However, by Def. 3.2, a sample $\mathbf{y}$ is in the *transferable set* $\mathcal{T}_\epsilon(\mathbf{x})$ of a sample $\mathbf{x}$ as long as there *exists* a practical solution to problem (3) (with some intrinsic quality gap $\epsilon$) that induces $\mathbf{y}$ to be misclassified as well. Thus, it is insufficient to decide whether or not a sample is in the transferable set of another sample according to merely one solution realization to problem (3). To address this, for each $\mathbf{x}_n^{(i)}$, we solve problem (3) *repeatedly* with random initialization. For each practical solution, we embed it to all elements in $\mathcal{D}_i \setminus \mathbf{x}_n^{(i)}$ and find those that are misclassified – these samples are included into the subset $\widehat{\mathcal{T}}_\epsilon(\mathbf{x}_n^{(i)})$ (line 9,

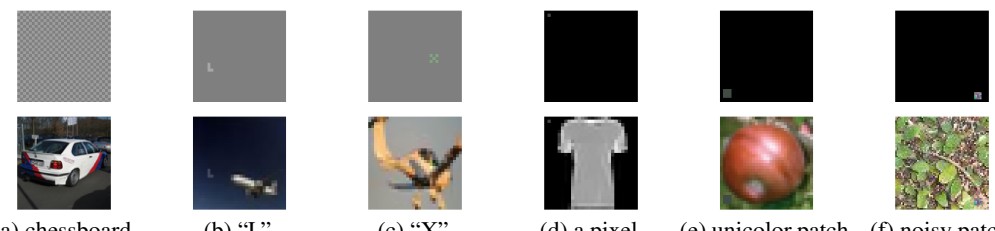

(a) chessboard  (b) "L"  (c) "X"  (d) a pixel  (e) unicolor patch  (f) noisy patch

Figure 1: Part of the BPs used in our experiments (others in Apdx. D.3) and images with these BPs embedded. (a)-(d) are additive perturbation BPs and (e)-(f) are patch replacement BPs. BP in (a) is amplified for visualization. Spatial locations for BPs in (b)-(f) are randomly selected for each attack.

Alg. 1). Such repetition stops when $\widehat{\mathcal{T}}_\epsilon(\mathbf{x}_n^{(i)})$ stays unchanged for some $\tau$ iterations. This procedure is summarized as the "while" loop in Alg. 1, which is guaranteed to converge in $(N_i - 1) \times \tau$ iterations. Finally, we obtain the estimated ET by averaging $p_n^{(i)}$ – the empirical estimation of $\mathrm{P}(\mathbf{Y} \in \mathcal{T}_\epsilon(\mathbf{X})|\mathbf{X} = \mathbf{x}_n^{(i)})$ – over all samples in $\mathcal{D}_i$.

Our detection framework has the following generalization capabilities. 1) **BP embedding mechanism.** As mentioned before, our detection rule can be adapted to BPs embedded by Eq. (2), with the same constant threshold $\frac{1}{2}$ on ET and only little modification to the while loop in Alg. 1 (see Apdx. C.3). 2) **BP reverse-engineering algorithm.** We do not limit our detector to any specific algorithm when solving problem (3) (line 8 in Alg. 1) – existing algorithms proposed by, e.g., Xiang et al. (2020); Carlini & Wagner (2017), and even future BP reverse-engineering algorithms can be used. 3) **Adoption for multi-class scenario.** When there are multiple classes, where each can possibly be a BA target, Alg. 1 can still be used for detection by, for each putative target class $t \in \mathcal{C}$, treating all the classes other than $t$ as a *super-class* and estimating ET on $\cup_{i \in \mathcal{C} \setminus t} \mathcal{D}_i$. In our experiments (next), we will evaluate our detection framework considering a variety of these extensions.

## 4 EXPERIMENTS

Our experiments involve six common benchmark image datasets with a variety of image size and color scale: CIFAR-10, CIFAR-100 Krizhevsky (2012), STL-10 Coates et al. (2011), TinyImageNet, FMNIST Xiao et al. (2017), MNIST Lecun et al. (1998). Details of these datasets are in Apdx. D.1.

### 4.1 MAIN EXPERIMENT: 2-CLASS, MULTI-ATTACK BA DETECTION USING ET STATISTIC

**Generating 2-class domains.** From CIFAR-10, we generate 45 different 2-class domains (for all 45 unordered class pairs of CIFAR-10). From *each* of the other five datasets, we generate 20 different random 2-class domains. More details are provided in Apdx. D.2.

**Attack configuration.** Like most related works, we mainly focus on classical BAs launched by poisoning the classifier's training set with a small set of samples embedded with a BP and labeled to some target class Gu et al. (2019). Effectiveness of our detector against another type of clean-label BA is shown in Apdx. E. For each 2-class domain generated from CIFAR-10, CIFAR-100, STL-10, and TinyImageNet, we create two *attack instances*, one for BA with additive perturbation BP embedded by Eq. (1), and the other for BA with patch replacement BP embedded by Eq. (2). For each 2-class domain generated from FMNIST and MNIST, we create one attack instance with additive perturbation BP[2]. For convenience, the six ensembles of attack instances with additive perturbation BP and for 2-class domains generated from CIFAR-10, CIFAR-100, STL-10, TinyImageNet, FM-NIST, and MNIST are denoted as $\mathbf{A_1}$-$\mathbf{A_6}$ respectively. The four ensembles of attack instances with patch replacement BP and for 2-class domains generated from CIFAR-10, CIFAR-100, STL-10, and TinyImageNet are denoted as $\mathbf{A_7}$-$\mathbf{A_{10}}$ respectively. The BPs used in our experiments include many popular ones in the BA literature – examples of some BPs and images embedded with them are shown in Fig. 1 (details for generating these BPs are deferred to Apdx. D.3).

We consider 2-class scenarios where *both classes can possibly be a BA target class.* For each attack instance in ensembles $A_1$, $A_2$, $A_3$, $A_7$, $A_8$, and $A_{10}$, we create two attacks each with one of the two classes being the BA target class. For each instance in ensembles $A_4$, $A_5$, $A_6$, and $A_9$, we create one attack with the second class being the BA target class. For each of these attacks, the BP is

---

[2]Images from these two datasets commonly have a large area of "black" background. Positively perturbing a few background pixels, which is a common practice to achieve a successful BA Chen et al. (2018), is *equivalent* to replacing these pixels with a gray patch using Eq. (2).

Table 1: Detection accuracy for RE-AP and RE-PR on attack ensembles $A_1$-$A_{10}$, and on clean ensembles $C_1$-$C_6$, using *the common threshold 1/2 on ET statistic*. "n/a" represents "not applicable".

| | $A_1$ | $A_2$ | $A_3$ | $A_4$ | $A_5$ | $A_6$ | $A_7$ | $A_8$ | $A_9$ | $A_{10}$ | $C_1$ | $C_2$ | $C_3$ | $C_4$ | $C_5$ | $C_6$ |
|---|---|---|---|---|---|---|---|---|---|---|---|---|---|---|---|---|
| RE-AP | 45/45 | 18/20 | 16/20 | 17/20 | 20/20 | 20/20 | n/a | n/a | n/a | n/a | 45/45 | 20/20 | 20/20 | 20/20 | 20/20 | 20/20 |
| RE-PR | n/a | n/a | n/a | n/a | n/a | n/a | 45/45 | 20/20 | 19/20 | 19/20 | 39/45 | 19/20 | 20/20 | 16/20 | 18/20 | 19/20 |

randomly selected from the candidate BPs (part of which are shown in Fig. 1), with the specified BP type for the ensemble which the attack is associated with. However, to avoid confusion for learning the backdoor mapping during training, for all 2-attack instances with additive perturbation BPs, we ensure that the BPs for the two attacks have different shapes (e.g., two "X" BPs are not allowed). Similarly, for all 2-attack instances where the two attacks both choose to use the unicolor patch BP (Fig. 1e), we ensure that the colors for the two BPs are significantly different. Other attack configurations, including e.g., the poisoning rate for each attack are detailed in Apdx. D.4.

**Training configurations.** We train one classifier for each attack instance using the poisoned training set. For each 2-class domain, we also train a clean classifier to evaluate false detections. We denote the six ensembles of clean instances for datasets CIFAR-10, CIFAR-100, STL-10, TinyImageNet, FMNIST, and MNIST as $C_1$-$C_6$ respectively. For classifier training, we consider a variety of DNN architectures (with two output neurons, one for each of the two classes). For 2-class domains associated with CIFAR-10, CIFAR-100, and STL-10, we use ResNet-18 He et al. (2016); for FMNIST, we use VGG-11 Simonyan & Zisserman (2015); for TinyImageNet, we use ResNet-34; and for MNIST, we use LeNet-5 Lecun et al. (1998). Other training configurations including the learning rate, the batch size, and the optimizer choice for each 2-class domain are detailed in Apdx. D.5. Moreover, all attacks for all the instances are successful with high *attack success rate* and negligible degradation in *clean test accuracy* – these two metrics are commonly used to evaluate the BA effectiveness Wang et al. (2020). As a defense paper, we defer these attack details to Apdx. D.5.

**Defense configurations.** As mentioned in Sec. 3.3, our detection framework can be easily generalized to incorporate a variety of algorithms for BP reverse-engineering to detect various types of BP with different embedding mechanisms. Here, we consider the algorithm proposed by Xiang et al. (2020) for estimating Additive Perturbation BPs (by solving (3)), and the algorithm proposed by Wang et al. (2019) for estimating Patch Replacement BPs (by solving (46) in Apdx. C.1). For convenience, we denote detection configurations with these two algorithms as **RE-AP** and **RE-PR**, respectively. Details for these two algorithms are both introduced in the original papers and reviewed in Apdx. D.6. Irrespective of the BP reverse-engineering algorithm, we use only 20 clean images per class (similar to most existing REDs) for detection. Finally, we set the "patience" parameter for determination of convergence in Alg. 1 to $\tau=4$ – this choice is independent of the presence of BA; a larger $\tau$ will not change the resulting ET much, but only increase the execution time.

**Detection performance (using the common ET threshold 1/2).** In practice, our detector with RE-AP and with RE-PR can be deployed *in parallel* to cover both additive perturbation BPs and patch replacement BPs. Here, for simplicity, we apply our detector with RE-AP to classifiers (with BAs using additive perturbation BP) in ensemble $A_1$-$A_6$, and apply our detector with RE-PR to classifiers (with BAs using patch replacement BP) in ensemble $A_7$-$A_{10}$. For each classifier in clean ensembles $C_1$-$C_6$, we apply our detector with both configurations. In Tab. 1, for each ensemble of attack instances, we report the fraction of classifiers such that the attack and all BA target classes are both successfully detected; for each ensemble of clean instances, we report the fraction of classifiers that are inferred to be not attacked. Given the large variety of classification domains, attack configurations, DNN architectures, and defense generalizations mentioned above, *using the common ET threshold 1/2*, we successfully detect most attacks with only very few false detections (see Tab. 1). More discussions regarding the detection performance are deferred to Apdx. K.

## 4.2 COMPARE ET WITH OTHER STATISTICS

The results for existing REDs applying to the attacks we created are neglected for brevity, since these REDs cannot detect BAs for 2-class domains by their design. However, we compare our ET statistic with some popular types of statistic used by existing REDs in terms of their potential for being used to **distinguish BA target classes from non-target classes**. The types of statistic for comparison include: 1) the $l_2$ norm of the estimated additive perturbation used by Xiang et al. (2020) (denoted by $\mathbf{L_2}$); 2) the $l_1$ norm of the estimated mask used by Wang et al. (2019) (denoted by $\mathbf{L_1}$); and 3) the (cosine) similarity between the BP estimated group-wise and the BP estimated for each sample in terms of classifier's internal layer representation Wang et al. (2020) (denoted by **CS**).

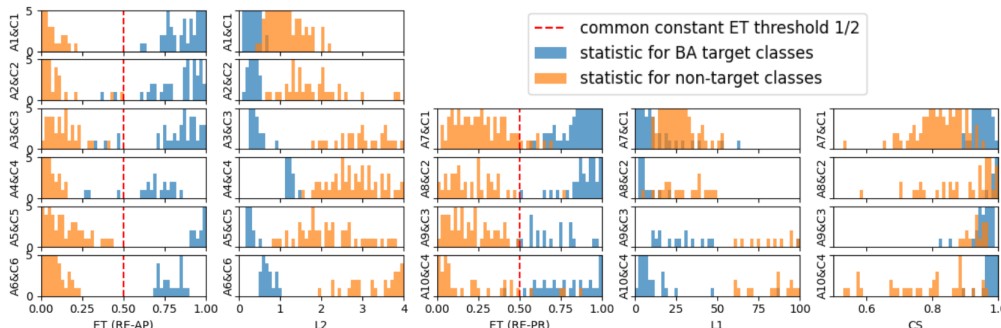

Figure 2: Comparison between our ET statistic (for both RE-AP and RE-PR configurations) and statistic types used by existing REDs ($L_1$, $L_2$, and CS). Only for ET, there is a common range for all 2-class domains for choosing a threshold to distinguish BA target classes (blue) from non-target classes (orange). Such common range also contains the constant threshold 1/2 (red dashed line).

For each type of statistic mentioned above and each benchmark dataset, we consider all classifiers (with and without BA) trained for all 2-class domains generated from the dataset, and plot a double histogram for statistics obtained for all BA target classes and all non-target classes across these classifiers respectively. For example, in the 1st column of Fig. 2, for our detector with RE-AP, we plot a double histogram for ET statistics obtained for all BA target classes and all non-target classes from all classifiers in each of $A_1\&C_1$, $A_2\&C_2$, $A_3\&C_3$, $A_4\&C_4$, $A_5\&C_5$, and $A_6\&C_6$. Note that all classifiers in, e.g., $A_1\&C_1$ are trained for 2-class domains generated from CIFAR-10. Here, for simplicity, for ET (with RE-AP) and $L_2$ (both designated for additive perturbation BP), we do not consider BA target classes with patch replacement BP (e.g. associated with classifiers in $A_7$-$A_{10}$); for ET (with RE-PR), $L_1$ and CS (designated for patch replacement BP), we do not consider BA target classes with additive perturbation BP (e.g. associated with classifiers in $A_1$-$A_6$).

Based on Fig. 2, for *all* types of statistics including our ET, statistics obtained for BA target classes are generally separable from statistics obtained for non-target classes for classifiers (with and without BA) trained for 2-class domains generated from the *same* benchmark dataset (using same training configurations including DNN architecture). But only for our ET (obtained by both RE-AP and RE-PR), there is a *common range* (irrespective of the classification domain, attack configurations, and training configurations) for choosing a detection threshold to effectively distinguish BA target classes from non-target classes for all instances; and such common range clearly includes the *constant threshold 1/2* (marked by red dashed lines in Fig. 2) derived mathematically in Sec. 3. By contrast, for both $L_1$ and $L_2$, a proper choice of detection threshold for distinguishing BA target classes from non-target classes is domain dependent. For example, in the 4th column of Fig. 2, the $l_1$ norm of masks estimated for both BA target classes and non-target classes for domains with larger image size (e.g. $96 \times 96$ for domains in $A_9\&C_3$ generated from STL-10) is commonly larger than for domains with smaller image size (e.g. $32 \times 32$ for domains in $A_7\&C_1$ generated from CIFAR-10). The CS statistic, on the other hand, not only relies on the domain, but also depends on the DNN architecture. More details regarding CS are deferred to Apdx. D.7.

In summary, all the above mentioned types of statistics are suitable for BA detection *if there is supervision from the same domain for choosing the best proper detection threshold*. However, in most practical scenarios where such supervision is not available, only our ET statistic with the common detection threshold 1/2 can still be used to achieve a good detection performance. Finally, in Fig. 3, we show the receiver operating characteristic (ROC) curves associated with Fig. 2 for each of ET, $L_1$, $L_2$, and SC – our ET statistic has clearly larger area under the ROC curve (very close to 1) than the other types of statistics.

### 4.3 EXPERIMENTAL VERIFICATION OF PROPERTY 3.1

We verify Property 3.1 by showing that, if a class is not a BA target class, for any two clean images from the other class (considering 2-class domains), the minimum additive perturbation required to induce both images to be misclassified has a *larger* norm than the minimum perturbation required for each of these two images to be misclassified. Here, we randomly choose one clean classifier from each of $C_1$-$C_6$. For each classifier, we randomly choose 50 *pairs* of clean images from a random class of the associated 2-class domain – these images are also used for detection in Sec. 4.1. For each pair of images, we apply the same RE-AP algorithm (i.e. the BP reverse-engineering algorithm

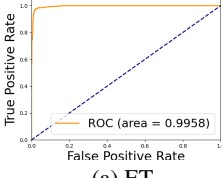 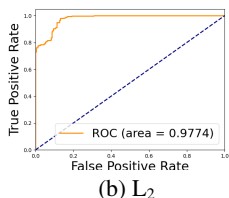 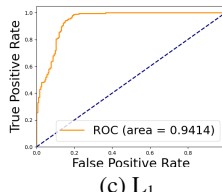 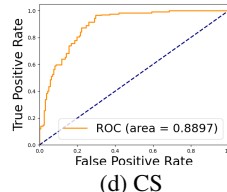

(a) ET        (b) $L_2$        (c) $L_1$        (d) CS

Figure 3: ROC curves for ET, $L_1$, $L_2$, and SC in distinguishing BA target classes from non-target classes for the large variety of classification domains and attack configurations considered in Fig. 2.

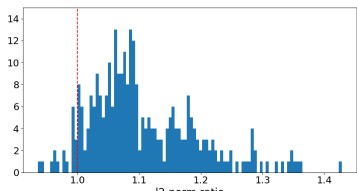

Figure 4: Histogram of $l_2$ norm ratio between pair-wise additive perturbation and maximum of the two sample-wise perturbations for each random image pair for clean classifiers.

Table 2: Maximum ET statistic over all classes for classifiers with one, two, and three attacks respectively, and a clean classifier, for CIFAR-10, CIFAR-100, and STL-10.

|  | 1 attack | 2 attacks | 3 attacks | clean |
|---|---|---|---|---|
| CIFAR-10 | 0.91 | 0.92 | 0.87 | 0 |
| CIFAR-100 | 0.95 | 0.99 | 0.99 | 0.27 |
| STL-10 | 0.65 | 0.83 | 0.77 | $4.3e^{-3}$ |

proposed by Xiang et al. (2020) for additive perturbation BPs) on the two images both *jointly* (to get a pair-wise common perturbation) and *separately* (to get two sample-wise perturbationsfor the two images respectively). Then, we divide the $l_2$ norm of the pair-wise perturbation by the *maximum* $l_2$ norm of the two sample-wise perturbations to get a ratio (which is more scale-insensitive than taking absolute difference). In Fig. 4, we plot the histogram of such ratio for all image pairs for all six classifiers – the ratio for most of the image pairs is greater than 1 (marked by the red dashed line in Fig. 4). For these pairs, very likely that the perturbation estimated for one sample cannot induce the other sample to be misclassified and vise versa (otherwise their expected ratio will likely be 1).

### 4.4 MULTI-CLASS, MULTI-ATTACK BA DETECTION USING ET STATISTIC

Since our detector inspects each class independently, it can also be used for BA detection for more general scenarios with *more than two classes* and *arbitrary number of attacks* (and BA target classes). For demonstration, for each of CIFAR-10, CIFAR-100, and STL-10, we create three attack instances with one, two, and three attacks, respectively (on the *original* domain). Additional experiments with aggregated results are deferred to Apdx. L due to space limitations. Here, we consider BAs with additive perturbation BPs for simplicity. The target class and the shape of BP for each attack are randomly selected. We train one classifier for each attack instance (thus, nine classifiers being attacked in total). More details for these attacks and training configurations are in Apdx. D.8. For each domain, we also train a clean classifier (without BA) for evaluating false detections.

Following the description at the end of Sec. 3, we apply the generalized Alg. 1 with RE-AP to these classifiers. For CIFAR-10 and STL-10, we use three clean images per class for detection; for CIFAR-100, we use only one clean image per class for detection. Other detection configurations including the *detection threshold 1/2* are the same as in Sec. 4.1. Since a classifier is deemed to be attacked if ET obtained for any class is greater than the threshold 1/2, for each classifier, we show the maximum ET over all the classes in Tab. 2. Clearly, the maximum ET is greater than 1/2 for all classifiers being attacked and less than 1/2 for all clean classifiers. Thus, our detection framework (with the same constant threshold on ET) is also applicable to multi-class, multi-attack scenarios.

## 5 CONCLUSIONS

We proposed the first BA detector for two-class, multi-attack scenarios without access to the classifier's training set or any supervision from clean reference classifiers trained for the same domain. Our detection framework is based on BP reverse-engineering and a novel ET statistic. Our ET statistic can be used to effectively distinguish BA target classes from non-target classes, with a particular, theoretical-grounded threshold value 1/2, irrespective of the classification domain, the DNN architecture, and the attack configurations. Our detection framework can also be generalized to incorporate a variety of BP reverse-engineering algorithms to address different BP types, and be applied to multi-class scenarios with arbitrary number of attacks.

ETHICS STATEMENT

The main purpose of this research is to understand the behavior of deep learning systems facing malicious activities, and enhance their safety level by unsupervised means. The backdoor attack considered this paper is well-known, with open-sourced implementation code. Thus, publication of this paper (with code released at `https://github.com/zhenxianglance/2ClassBADetection`) will be beneficial to the community in defending backdoor attacks.

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

# A  PROOF OF THEOREMS IN THE MAIN PAPER

## A.1  PROOF OF THEOREM 3.1

For random variables $\mathbf{X}$ and $\mathbf{Y}$ i.i.d. following distribution $P_i$, for any realization $\mathbf{x}$ of $\mathbf{X}$, we have

$$\mathrm{P}(\mathbf{Y} \in \mathcal{T}_\epsilon(\mathbf{x})) = \mathrm{P}(\mathbf{Y} \in \mathcal{T}_\epsilon(\mathbf{X})|\mathbf{X} = \mathbf{x}) = \mathbb{E}\big[\mathbb{1}(\mathbf{Y} \in \mathcal{T}_\epsilon(\mathbf{X}))\big|\mathbf{X} = \mathbf{x}\big] \tag{8}$$

Thus, ET, the left hand side of Eq. (6), can be written as

$$\mathbb{E}\big[\mathrm{P}(\mathbf{Y} \in \mathcal{T}_\epsilon(\mathbf{X})|\mathbf{X})\big] = \mathbb{E}\big[\mathbb{1}(\mathbf{Y} \in \mathcal{T}_\epsilon(\mathbf{X}))\big] = \mathrm{P}(\mathbf{Y} \in \mathcal{T}_\epsilon(\mathbf{X})) \tag{9}$$

Based on the inclusion-exclusion rule

$$\mathrm{P}(\mathbf{Y} \in \mathcal{T}_\epsilon(\mathbf{X})) + \mathrm{P}(\mathbf{X} \in \mathcal{T}_\epsilon(\mathbf{Y})) = 1 + \mathrm{P}_{\mathrm{MT},i} - \mathrm{P}_{\mathrm{NT},i} \tag{10}$$

Since, $\mathbf{X}$ and $\mathbf{Y}$ are i.i.d. random variables, $\mathrm{P}(\mathbf{Y} \in \mathcal{T}_\epsilon(\mathbf{X})) = \mathrm{P}(\mathbf{X} \in \mathcal{T}_\epsilon(\mathbf{Y}))$; thus, we get Eq. (6).

## A.2  PROOF OF THEOREM 3.2

Step 1: We show that for any $i \in \mathcal{C}$ and $\mathbf{X}$, $\mathbf{Y}$ i.i.d. following distribution $P_i$, $\mathbf{X} \in \mathcal{T}_\epsilon(\mathbf{Y})$ and $\mathbf{Y} \in \mathcal{T}_\epsilon(\mathbf{X})$ *if and only if* $\mathcal{V}_\epsilon(\mathbf{X}) \cap \mathcal{V}_\epsilon(\mathbf{Y}) \neq \emptyset$.

(a) If $\mathcal{V}_\epsilon(\mathbf{X}) \cap \mathcal{V}_\epsilon(\mathbf{Y}) \neq \emptyset$, there exists $\mathbf{v}$ such that $\mathbf{v} \in \mathcal{V}_\epsilon(\mathbf{X})$ and $\mathbf{v} \in \mathcal{V}_\epsilon(\mathbf{Y})$. By Definition 3.1,

$$\mathbf{v} \in \mathcal{V}_\epsilon(\mathbf{X}) \Rightarrow f(\mathbf{X} + \mathbf{v}) \neq f(\mathbf{X}) \tag{11}$$
$$\mathbf{v} \in \mathcal{V}_\epsilon(\mathbf{Y}) \Rightarrow f(\mathbf{Y} + \mathbf{v}) \neq f(\mathbf{Y}) \tag{12}$$

Then, by Definition 3.2, the existence of such $\mathbf{v}$ yields $\mathbf{X} \in \mathcal{T}_\epsilon(\mathbf{Y})$ and $\mathbf{Y} \in \mathcal{T}_\epsilon(\mathbf{X})$.

(b) We prove the "only if" part by contradiction. Given $\mathbf{X} \in \mathcal{T}_\epsilon(\mathbf{Y})$ and $\mathbf{Y} \in \mathcal{T}_\epsilon(\mathbf{X})$, suppose $\mathcal{V}_\epsilon(\mathbf{X}) \cap \mathcal{V}_\epsilon(\mathbf{Y}) = \emptyset$. By Definition 3.2, if $\mathbf{Y} \in \mathcal{T}_\epsilon(\mathbf{X})$, there exists $\mathbf{v} \in \mathcal{V}_\epsilon(\mathbf{X})$ such that $f(\mathbf{Y} + \mathbf{v}) \neq f(\mathbf{Y})$. Since $\mathcal{V}_\epsilon(\mathbf{X}) \cap \mathcal{V}_\epsilon(\mathbf{Y}) = \emptyset$, $\mathbf{v} \notin \mathcal{V}_\epsilon(\mathbf{Y})$; hence, by Definition 3.1

$$||\mathbf{v}||_2 - ||\mathbf{v}^*(\mathbf{Y})||_2 > \epsilon. \tag{13}$$

Similarly, there exists $\mathbf{v}' \in \mathcal{V}_\epsilon(\mathbf{Y})$ such that $f(\mathbf{X} + \mathbf{v}') \neq f(\mathbf{X})$ and $\mathbf{v}' \notin \mathcal{V}_\epsilon(\mathbf{X})$; hence

$$||\mathbf{v}'||_2 - ||\mathbf{v}^*(\mathbf{X})||_2 > \epsilon. \tag{14}$$

By Definition 3.1, for all $\mathbf{u} \in \mathcal{V}_\epsilon(\mathbf{Y})$, $||\mathbf{u}||_2 - ||\mathbf{v}^*(\mathbf{Y})||_2 < \epsilon$. Thus, we have

$$||\mathbf{v}||_2 - ||\mathbf{v}^*(\mathbf{Y})||_2 > ||\mathbf{u}||_2 - ||\mathbf{v}^*(\mathbf{Y})||_2 \tag{15}$$

and accordingly $||\mathbf{v}||_2 > ||\mathbf{u}||_2$ for all $\mathbf{u} \in \mathcal{V}_\epsilon(\mathbf{Y})$. Similarly, for all $\mathbf{u}' \in \mathcal{V}_\epsilon(\mathbf{X})$, we have

$$||\mathbf{v}'||_2 - ||\mathbf{v}^*(\mathbf{X})||_2 > ||\mathbf{u}'||_2 - ||\mathbf{v}^*(\mathbf{X})||_2 \tag{16}$$

and thus, $||\mathbf{v}'||_2 > ||\mathbf{u}'||_2$ for all $\mathbf{u}' \in \mathcal{V}_\epsilon(\mathbf{X})$. Clearly, there is a contradiction since there cannot exist an element in one set having a larger norm than all elements in another set while vise versa; and therefore, $\mathcal{V}_\epsilon(\mathbf{X}) \cap \mathcal{V}_\epsilon(\mathbf{Y}) \neq \emptyset$.

Step 2: We show that the upper bound for $\mathrm{P}_{\mathrm{MT},i}$ goes to 0 when $\epsilon \to 0$. Note that $\epsilon$ is the "quality gap" between the practical solution and the optimal solution.

Note that by Definition 3.1, $\mathcal{V}_\epsilon(\mathbf{X}) \cap \mathcal{V}_\epsilon(\mathbf{Y}) \neq \emptyset$ *only if* $\big|||\mathbf{v}^*(\mathbf{X})||_2 - ||\mathbf{v}^*(\mathbf{Y})||_2\big| \leq \epsilon$; also based on Step 1:

$$\mathrm{P}_{\mathrm{MT},i} = \mathrm{P}(\mathcal{V}_\epsilon(\mathbf{X}) \cap \mathcal{V}_\epsilon(\mathbf{Y}) \neq \emptyset) \leq \mathrm{P}(\big|||\mathbf{v}^*(\mathbf{X})||_2 - ||\mathbf{v}^*(\mathbf{Y})||_2\big| \leq \epsilon). \tag{17}$$

For $\mathbf{X}$ following continuous distribution $P_i$, $||\mathbf{v}^*(\mathbf{X})||_2$ also follows some continuous distribution[3] on $(0, +\infty)$. Since $\mathbf{X}$ and $\mathbf{Y}$ are i.i.d., the right hand side of Eq. (17) goes to 0 as $\epsilon \to 0$.

---

[3]One can construct very extreme cases such that $\mathrm{P}(||\mathbf{v}^*(\mathbf{X})||_2 = d) > 0$ for some constant $d > 0$. In other words, there is a set of samples from class $i$ with non-negligible probability that are *equal distant* to the decision boundary. However, the probability for these cases is zero for practical domains and highly non-linear classifiers.

## A.3 PROOF OF THEOREM 3.3

For any $\mathbf{X} \sim P_i$, since $\mathbf{v}_0$ satisfies $f(\mathbf{X} + \mathbf{v}_0) \neq f(\mathbf{X})$ and

$$||\mathbf{v}_0||_2 - ||\mathbf{v}^*(\mathbf{X})||_2 < ||\mathbf{v}_0||_2 \leq \epsilon, \tag{18}$$

$\mathbf{v}_0 \in \mathcal{V}_\epsilon(\mathbf{X})$ by Definition 3.1. Thus, $\mathbf{v}_0 \in \mathcal{V}_\epsilon(\mathbf{X}) \cap \mathcal{V}_\epsilon(\mathbf{Y})$ for $\mathbf{X}$ and $\mathbf{Y}$ i.i.d. following $P_i$; and

$$P(\mathcal{V}_\epsilon(\mathbf{X}) \cap \mathcal{V}_\epsilon(\mathbf{Y}) \neq \emptyset) = 1. \tag{19}$$

Thus, based on Step 1 (and Eq. (17)) in Apdx A.2, we have $P_{\mathrm{MT}} = 1$. Since $P_{\mathrm{MT}} + P_{\mathrm{NT}} \leq 1$, Eq. (6) can be written as

$$\mathrm{ET}_{i,\epsilon} \geq \frac{1}{2} + \frac{1}{2}(P_{\mathrm{MT}} - 1 + P_{\mathrm{MT}}) = P_{\mathrm{MT}} \tag{20}$$

Then we have

$$\mathrm{ET}_{i,\epsilon} = 1. \tag{21}$$

# B ANALYSIS ON A SIMPLIFIED CLASSIFICATION PROBLEM

In this section, we consider a simplified analogue to practical 2-class classification problems. Although assumptions are imposed for simplicity, we still keep the problem relatively general by allowing freedom on, e.g., the sample distribution in their latent space. With the simplification, we are able to analytically derive the condition for a sample belonging to the transferable set of another sample from the same class. Based on this, we show that $\frac{1}{2}$ is the *supremum* of the ET statistic for this problem when there is no attack. Note that we will abuse some notations that appeared in the main paper; *the notations in this section are all self-contained*.

## B.1 PROBLEM SETTINGS

We consider sample space $\mathbb{R}^n$ with an orthonormal basis $\{\boldsymbol{\alpha}_1, \cdots, \boldsymbol{\alpha}_d, \boldsymbol{\beta}_1, \cdots, \boldsymbol{\beta}_{n-d}\}$. We assume that samples from the two classes are distributed on sub-spaces $V_0 = \{\mathbf{Ac}|\mathbf{c} \in \mathbb{R}^d\}$ and $V_1 = \{\mathbf{Be}|\mathbf{e} \in \mathbb{R}^{n-d}\}$ respectively, with $\mathbf{A} = [\boldsymbol{\alpha}_1, \cdots, \boldsymbol{\alpha}_d]$ and $\mathbf{B} = [\boldsymbol{\beta}_1, \cdots, \boldsymbol{\beta}_{n-d}]$. By the definition of $V_0$ and $V_1$, we have also defined the *latent spaces*, i.e. $\mathbb{R}^d$ and $\mathbb{R}^{n-d}$, for the two classes. Here, we do not constrain the form or parameters of the distributions for the two classes in both the sample space $\mathbb{R}^n$ and latent spaces, as long as the distributions are continuous. Such an analogue may be corresponding to some simple classification domains in practice. Moreover, the latent space may be corresponding to an internal layer space of some deep neural network (DNN) classifier. For example, for a typical ReLU DNN classifier, it is possible that a subset of nodes in the penultimate layer are mainly activated for one class, while another subset of nodes are mainly activated for the other class.

For this simplified domain, we consider a nearest prototype classifier that is capable of classifying the two classes perfectly for *any* continuous sample distributions. To achieve this, any point $\mathbf{x} \in \mathbb{R}^n$ on the decision boundary of the classifier should have equal distance to $V_0$ and $V_1$, i.e.:

$$||\mathbf{AA}^T\mathbf{x} - \mathbf{x}||_2 = ||\mathbf{BB}^T\mathbf{x} - \mathbf{x}||_2 \tag{22}$$

By expanding both sides of Eq. (22) and rearranging terms (using the fact that $\mathbf{A}^T\mathbf{A} = \mathbf{I}$ and $\mathbf{B}^T\mathbf{B} = \mathbf{I}$), we obtain the decision boundary of the classifier, which is $\{\mathbf{x} \in \mathbb{R}^n | \mathbf{x}(\mathbf{AA}^T - \mathbf{BB}^T)\mathbf{x} = 0\}$. In other words, the classifier $f : \mathbb{R}^n \to \{0, 1\}$ is defined by

$$f(\mathbf{x}) = \begin{cases} 0 & \mathbf{x}(\mathbf{AA}^T - \mathbf{BB}^T)\mathbf{x} > 0 \\ 1 & \mathbf{x}(\mathbf{AA}^T - \mathbf{BB}^T)\mathbf{x} \leq 0 \end{cases} \tag{23}$$

Note that this classifier is not necessarily linear; and the region for each class may not be convex.

## B.2 DERIVATION OF TRANSFER CONDITION

We derive the condition that one sample belongs to the transferable set of another sample from the same class. Since the two classes are symmetric, we focus on class 0 for brevity. Moreover, in this section, we refer to samples by their latent space representation; i.e., instead of "a sample $\mathbf{x} \in \mathbb{R}^n$ from class 0", we use "a sample $\mathbf{c} \in \mathbb{R}^d$".

First, we present the following modified definition of the transferable set (compared with Def. 3.2 in the main paper) using latent space representation.

**Definition B.1. (Transferable set in latent space)** The transferable set for any sample $\mathbf{c} \in \mathbb{R}^d$ from class 0 is defined by

$$\mathcal{T}(\mathbf{c}) = \{\mathbf{c}' \in \mathbb{R}^d \big| f(\mathbf{Ac}') = f(\mathbf{Ac}), f(\mathbf{Ac}' + \mathbf{v}^*(\mathbf{c})) \neq f(\mathbf{Ac}')\}, \tag{24}$$

where $\mathbf{v}^*(\mathbf{c})$ is the optimal solution to

$$\underset{\mathbf{v} \in \mathbb{R}^n}{\text{minimize}} \, ||\mathbf{v}||_2 \qquad \text{subject to } f(\mathbf{Ac} + \mathbf{v}) \neq f(\mathbf{Ac}). \tag{25}$$

Note that the above definition is in similar form to Def. 3.2 in the main paper, though here, the quality to the solution to problem (25) is no longer considered. This is because, different with problem (3) in the main paper (which is the prerequisite of Def. 3.2), problem (25) here can be solved analytically, yielding a close form solution instead of a solution set with some intrinsic quality bound $\epsilon$. Accordingly, we present the following theorem, which gives the condition for one sample belonging to the transferable set of another sample from the same class.

**Theorem B.1.** For any $\mathbf{c}, \mathbf{c}' \in \mathbb{R}^d$, $\mathbf{c}' \in \mathcal{T}(\mathbf{c})$ if and only if $||\mathbf{c}' - \mathbf{c}/2||_2 \leq ||\mathbf{c}/2||_2$.

*Proof.* First, we derive the solution to problem (25). Note that $\mathbf{v} \in \mathbb{R}^n$ can be decomposed (using the orthonormal basis specified by $\mathbf{A}$ and $\mathbf{B}$) as $\mathbf{v} = \mathbf{Av}_a + \mathbf{Bv}_b$ with $\mathbf{v}_a \in \mathbb{R}^d$ and $\mathbf{v}_b \in \mathbb{R}^{n-d}$. We substitute this decomposition into the constraint of problem (25). In words, the constraint means that $\mathbf{v}$ induces $\mathbf{Ac}$ to be (mis)classified to class 1; thus, according to the expression of the classifier in Eq. (23), the constraint can be written as

$$(\mathbf{Ac} + \mathbf{Av}_a + \mathbf{Bv}_b)^T(\mathbf{AA}^T - \mathbf{BB}^T)(\mathbf{Ac} + \mathbf{Av}_a + \mathbf{Bv}_b) \leq 0. \tag{26}$$

Expanding the left hand side of the above inequality and using the fact that $\mathbf{A}^T\mathbf{B} = 0$ for simplification, we get a much simpler expression of the constraint:

$$||\mathbf{v}_a + \mathbf{c}||_2 \leq ||\mathbf{v}_b||_2. \tag{27}$$

Thus, a lower bound of the (square of the) objective to be minimized in problem (25) can be derived as

$$||\mathbf{v}||_2^2 = ||\mathbf{Av}_a + \mathbf{Bv}_b||_2^2 = ||\mathbf{v}_a||_2^2 + ||\mathbf{v}_b||_2^2 \geq ||\mathbf{v}_a||_2^2 + ||\mathbf{v}_a + \mathbf{c}||_2^2, \tag{28}$$

with equality holds if and only if $||\mathbf{v}_b||_2 = ||\mathbf{v}_a + \mathbf{c}||$, i.e. $\mathbf{Ac} + \mathbf{v}$ on the decision boundary of the classifier. Note that the right hand side of the inequality above is minimized when $\mathbf{v}_a = -\frac{\mathbf{c}}{2}$. Then, the optimal solution $\mathbf{v}^*(\mathbf{c}) = \mathbf{Av}_a^*(\mathbf{c}) + \mathbf{Bv}_b^*(\mathbf{c})$ to problem (25) satisfies:

$$\begin{cases} \mathbf{v}_a^*(\mathbf{c}) = -\frac{\mathbf{c}}{2}, \\ ||\mathbf{v}_b^*(\mathbf{c})||_2 = \frac{||\mathbf{c}||_2}{2}. \end{cases} \tag{29}$$

Next, for any $\mathbf{c}, \mathbf{c}' \in \mathbb{R}^d$, we derive the condition for $\mathbf{c}' \in \mathcal{T}(\mathbf{c})$. Since $f(\mathbf{Ac}') = f(\mathbf{Ac})$ is already satisfied, by Def. B.1, $\mathbf{c}' \in \mathcal{T}(\mathbf{c})$ if and only if $f(\mathbf{Ac}' + \mathbf{v}^*(\mathbf{c})) \neq f(\mathbf{Ac}')$, which is equivalent to (by Eq. (23))

$$(\mathbf{Ac}' + \mathbf{v}^*(\mathbf{c}))^T(\mathbf{AA}^T - \mathbf{BB}^T)(\mathbf{Ac}' + \mathbf{v}^*(\mathbf{c})) \leq 0. \tag{30}$$

Using the decomposition of $\mathbf{v}^*(\mathbf{c})$, expanding and rearranging terms on the left hand side of the above, we obtain

$$\mathbf{c}'^T\mathbf{c}' + \mathbf{v}_a^*(\mathbf{c})^T\mathbf{v}_a^*(\mathbf{c}) - \mathbf{v}_b^*(\mathbf{c})^T\mathbf{v}_b^*(\mathbf{c}) + 2\mathbf{c}'^T\mathbf{v}_a^*(\mathbf{c}) \leq 0. \tag{31}$$

With the optimal solution in Eq. (29) substituted in, we get

$$\mathbf{c}'^T(\mathbf{c}' - \mathbf{c}) \leq 0, \tag{32}$$

or, equivalently

$$||\mathbf{c}' - \mathbf{c}/2||_2 \leq ||\mathbf{c}/2||_2. \tag{33}$$

$\square$

### B.3 UPPER BOUND ON ET STATISTIC

In this section, we show that $\frac{1}{2}$ is the *minimum upper bound* on the ET statistic for this problem when there is no BA. To do so, we first present a definition of ET statistic modified from Def 3.3 in the main paper, merely in adaption to the latent space representation used here (see Def. B.2). Then, we show the tightness of the bound by giving a concrete example where ET equals $\frac{1}{2}$ in Lem. B.1. Finally, we prove that the ET statistic cannot be greater than $\frac{1}{2}$.

**Definition B.2.** For i.i.d. random samples $\mathbf{C}$ and $\mathbf{C}'$ following some continuous distribution $G$ on $\mathbb{R}^d$, the ET statistic is defined by

$$\text{ET} = \mathbb{E}_{\mathbf{C}\sim G}\big[\text{P}(\mathbf{C}' \in \mathcal{T}(\mathbf{C})|\mathbf{C})\big] \tag{34}$$

**Lemma B.1.** For latent space $\mathbb{R}^d$ with dimension $d = 1$ and distribution $G$ continuous on $\mathbb{R}$, the ET statistic satisfies

$$\frac{1}{4} \le \text{ET} \le \frac{1}{2}, \tag{35}$$

where $\text{ET} = \frac{1}{2}$ if and only if $G(0) = 0$ or $G(0) = 1$.

*Proof.* Based on Thm. B.1, for latent space dimension $d = 1$ and two scalar[4] i.i.d. random samples $C$ and $C'$ with continuous distribution $G$, $C' \in \mathcal{T}(C)$ if and only if $|C' - C/2| \le |C/2|$. Thus, by Def. B.2, we have

$$
\begin{aligned}
\text{ET}_{(d=1)} &= \mathbb{E}_{C\sim G}[G(\frac{C}{2} + \frac{|C|}{2}) - G(\frac{C}{2} - \frac{|C|}{2})] \\
&= \int_{-\infty}^{\infty} [G(\frac{c}{2} + \frac{|c|}{2}) - G(\frac{c}{2} - \frac{|c|}{2})]g(c)dc \\
&= \int_{-\infty}^{0} [G(0) - G(c)]g(c)dc + \int_{0}^{\infty} [G(c) - G(0)]g(c)dc \\
&= \int_{0}^{G(0)} [G(0) - G(c)]dG(c) + \int_{G(0)}^{1} [G(c) - G(0)]dG(c) \\
&= G(0)^2 - \frac{1}{2}G(0)^2 + \frac{1}{2}[1 - G(0)^2] - G(0)[1 - G(0)] \\
&= \frac{1}{2} - G(0) + G(0)^2,
\end{aligned}
\tag{36}
$$

where $g(\cdot)$ is the density function of distribution $G$. Note that the last line of Eq. (36) is strictly in the interval $[\frac{1}{4}, \frac{1}{2}]$ for $G(\cdot)$ in range $[0, 1]$. The upper bound of ET when $d = 1$ is $\frac{1}{2}$, which is achieved if and only if $G(0) = 0$ or $G(0) = 1$. $\qquad\square$

**Theorem B.2.** For arbitrary $d \in \mathbb{Z}_+$ and continuous distribution $G$ on $\mathbb{R}^d$

$$\sup_{d,G} \quad \text{ET} = \frac{1}{2}. \tag{37}$$

*Proof.* By Lem. B.1, there exists $d = 1$ and $G(0) = 0$ or $G(0) = 1$, such that $\text{ET} = \frac{1}{2}$. Here, we only need to show that $\text{ET} \le \frac{1}{2}$ for any $d$ and $G$. Again, by the definition of ET (Def. B.2) and Thm. B.1, we can write ET as

$$\text{ET} = \mathbb{E}_{\mathbf{C}\sim G}\Big[\text{P}(||\mathbf{C}' - \frac{\mathbf{C}}{2}||_2 \le ||\frac{\mathbf{C}}{2}||_2)|\mathbf{C})\Big], \tag{38}$$

for $\mathbf{C}$ and $\mathbf{C}'$ i.i.d. following distribution $G$. Similar to our proof of Thm. 3.1 in Sec. A.1

$$\text{P}(||\mathbf{C}' - \frac{\mathbf{c}}{2}||_2 \le ||\frac{\mathbf{c}}{2}||_2) = \mathbb{E}_{\mathbf{C}'\sim G}\Big[\mathbb{1}\{||\mathbf{C}' - \frac{\mathbf{C}}{2}||_2 \le ||\frac{\mathbf{C}}{2}||_2\}\Big|\mathbf{C} = \mathbf{c}\Big]. \tag{39}$$

Thus, based on Eq. (38) and (39), ET can be written as

$$\text{ET} = \mathbb{E}_{\mathbf{C},\mathbf{C}'\sim G}\Big[\mathbb{1}\{||\mathbf{C}' - \frac{\mathbf{C}}{2}||_2 \le ||\frac{\mathbf{C}}{2}||_2)\}\Big] = \text{P}(||\mathbf{C}' - \frac{\mathbf{C}}{2}||_2 \le ||\frac{\mathbf{C}}{2}||_2) \tag{40}$$

---

[4]We do not use bold $\mathbf{C}$ but $C$ instead, since it is given as a scalar random variable.

Since $\mathbf{C}$ and $\mathbf{C}'$ are i.i.d,

$$P(||\mathbf{C}' - \frac{\mathbf{C}}{2}||_2 \le ||\frac{\mathbf{C}}{2}||_2) = P(||\mathbf{C} - \frac{\mathbf{C}'}{2}||_2 \le ||\frac{\mathbf{C}'}{2}||_2) \tag{41}$$

Also note that for continuous distribution

$$\begin{aligned}
&P(||\mathbf{C}' - \frac{\mathbf{C}}{2}||_2 \le ||\frac{\mathbf{C}}{2}||_2, ||\mathbf{C} - \frac{\mathbf{C}'}{2}||_2 \le ||\frac{\mathbf{C}'}{2}||_2) \\
&\le P(||\mathbf{C}' - \frac{\mathbf{C}}{2}||_2^2 + ||\mathbf{C} - \frac{\mathbf{C}'}{2}||_2^2 \le ||\frac{\mathbf{C}}{2}||_2^2 + ||\frac{\mathbf{C}'}{2}||_2^2) \\
&= P(||\mathbf{C} - \mathbf{C}'||_2^2 \le 0) \\
&= 0
\end{aligned} \tag{42}$$

Then, by the inclusion-exclusion rule

$$\begin{aligned}
&P(||\mathbf{C}' - \frac{\mathbf{C}}{2}||_2 \le ||\frac{\mathbf{C}}{2}||_2) + P(||\mathbf{C} - \frac{\mathbf{C}'}{2}||_2 \le ||\frac{\mathbf{C}'}{2}||_2) \\
&- P(||\mathbf{C}' - \frac{\mathbf{C}}{2}||_2 \le ||\frac{\mathbf{C}}{2}||_2, ||\mathbf{C} - \frac{\mathbf{C}'}{2}||_2 \le ||\frac{\mathbf{C}'}{2}||_2) \le 1,
\end{aligned} \tag{43}$$

Substituting Eq. (41) and (42) into Eq. (43), we have

$$P(||\mathbf{C}' - \frac{\mathbf{C}}{2}||_2 \le ||\frac{\mathbf{C}}{2}||_2) \le \frac{1}{2}, \tag{44}$$

thus, by Eq. (40)

$$\text{ET} \le \frac{1}{2} \tag{45}$$

which finishes the proof. □

## C   USING ET TO DETECT BA WITH PATCH REPLACEMENT BP

In the main paper, we have described our detection framework in details considering additive perturbation BPs embedded by Eq. (1). In fact, our detection framework is *not* specific to any particular backdoor embedding mechanism. In this section, we repeat our derivation and analysis in Sec. 3 by considering patch replacement BPs embedded by Eq. (2). Basically, the definitions, theorems and algorithms presented in this section are matched with those in the main paper – *they are under the same framework which is generally independent of the backdoor pattern embedding mechanism.*

Like the organization of Sec. 3, in this section, we first introduce several definitions related to ET but for patch replacement BPs – these definitions are similar to those in the main paper and are customized to patch replacement BPs. Especially, the ET statistic is defined in the same fashion as in Def. 3.3 of the main paper. Then, we show that the same constant detection threshold $\frac{1}{2}$ on the ET statistic for detecting BAs with with additive perturbation BPs can be used for detecting BAs with patch replacement BPs as well. Finally, we present the detailed procedure of our detection for patch replacement BPs (as the counterpart of Alg. 1 of the main paper). Again, this section can be viewed as an independent section, where *the notations are self-contained.*

### C.1   DEFINITION OF ET FOR PATCH REPLACEMENT BP

Consider the same classifier $f : \mathcal{X} \to \mathcal{C}$ to be inspected (as in the main paper) with the same label space $\mathcal{C} = \{0, 1\}$ and continuous sample distribution $P_i$ on $\mathcal{X}$ for class $i \in \mathcal{C}$. For any sample $\mathbf{x}$ from any class, the optimal solution to

$$\underset{\mathbf{s}=\{\mathbf{m},\mathbf{u}\}}{\text{minimize}} ||\mathbf{m}||_1 \qquad \text{subject to } f(\boldsymbol{\Delta}(\mathbf{x}; \mathbf{m}, \mathbf{u})) \ne f(\mathbf{x}) \tag{46}$$

is defined as $\mathbf{s}^*(\mathbf{x}) = \{\mathbf{m}^*(\mathbf{x}), \mathbf{u}^*(\mathbf{x})\}$, where $\boldsymbol{\Delta}(\mathbf{x}; \mathbf{m}, \mathbf{u}) = (1-\mathbf{m})\odot\mathbf{x}+\mathbf{m}\odot\mathbf{u}$ is an alternative expression to the patch replacement embedding formula in Eq. (2) (for notational convenience only). Again, $\boldsymbol{m} \in \mathcal{M}$ is the binary mask and $\mathbf{u} \in \mathcal{X}$ is the patch for replacement.

Existing methods for solving problem (46) including the one proposed by Wang et al. (2019), where, again, the practical solutions are usually sub-optimal. Thus, similar to Def. 3.1 in the main paper, we present the following definition in adaption to patch replacement BPs.

**Definition C.1. ($\epsilon$-solution set for patch replacement BPs)** For any sample $\mathbf{x}$ from any class, regardless of the method being used, the $\epsilon$-solution set to problem (46), is defined by

$$\mathcal{S}_\epsilon(\mathbf{x}) \triangleq \{\{\mathbf{m}, \mathbf{u}\} \big| \|\mathbf{m}\|_1 - \|\mathbf{m}^*(\mathbf{x})\|_1 \leq \epsilon, f(\mathbf{\Delta}(\mathbf{x}; \mathbf{m}, \mathbf{u})) \neq f(\mathbf{x})\}, \tag{47}$$

where $\epsilon > 0$ is the "quality" bound of the solutions, which is usually small for existing methods.

Similar to Def. 3.2 in the main paper, we present the following definition (with abused notation) for the transferable set for patch replacement BPs.

**Definition C.2. (Transferable set for patch replacement BPs)** The transferable set for any sample $\mathbf{x}$ and $\epsilon > 0$ is defined by

$$\mathcal{T}_\epsilon(\mathbf{x}) \triangleq \{\mathbf{y} \in \mathcal{X} \big| f(\mathbf{y}) = f(\mathbf{x}), \exists\{\mathbf{m}, \mathbf{u}\} \in \mathcal{S}_\epsilon(\mathbf{x}) \, \text{s.t.} \, f(\mathbf{\Delta}(\mathbf{y}; \mathbf{m}, \mathbf{u})) \neq f(\mathbf{y})\}. \tag{48}$$

Finally, we present the following definition of the ET statistic for patch replacement BPs, which looks *exactly the same* as Def. 3.3 in the main paper. However, here, the definition of the transferable set has been customized for patch replacement BPs. Even though, the similarity between these definitions and their counterparts in the main paper has already highlighted the generalization capability of our detection framework.

**Definition C.3. (ET statistic for patch replacement BPs)** For any class $i \in \mathcal{C} = \{0, 1\}$ and $\epsilon > 0$, considering i.i.d. random samples $\mathbf{X}, \mathbf{Y} \sim P_i$, the ET statistic for class $i$ is defined by $\text{ET}_{i,\epsilon} \triangleq \mathbb{E}\big[\text{P}(\mathbf{Y} \in \mathcal{T}_\epsilon(\mathbf{X})|\mathbf{X})\big]$.

## C.2 Detecting BA with Patch Replacement BP Using ET

For the ET statistic for patch replacement BPs defined above, the same constant detection threshold $\frac{1}{2}$ can be used for distinguish BA target classes from non-target classes. The connection between the ET statistic for patch replacement BPs and the constant threshold $\frac{1}{2}$ is established by the same Thm. 3.1 in Sec. 3.2 of the main paper with the same proof in Apdx. A.1. Thus, these details are not included here for brevity. Note that $\text{P}_{\text{MT},i}$ and $\text{P}_{\text{NT},i}$ for class $i$ are defined in the same way as in Thm. 3.1, though the transferable set $\mathcal{T}_\epsilon(\mathbf{x})$ is customized for patch replacement BPs in the current section – this is the main reason why we abuse the notation for the transferable set for patch replacement BPs in Def. C.2.

In the following, like in Sec. 3.2 of the main paper, we discuss the non-attack case and the attack case respectively. Readers should notice that the theorems (and the associated proofs) and discussions are similar to those in the main paper. Such similarity further highlight the generalization capability of our detection framework.

**Non-attack case.** Property 3.1 from the main paper is also applicable here. That is, if class $(1 - i)$ where $i \in \mathcal{C} = \{0, 1\}$ is not a BA target class, $\text{P}_{\text{NT},i}$ for class $i$ will likely be larger than $\frac{1}{2}$.

Similar to our verification of Property 3.1 for additive perturbation BPs in Sec. 4.3 of the main paper, we verify Property 3.1 for patch replacement BPs in Apdx. F. Using similar protocol as in Sec. 4.3 of the main paper, we show that the common mask with the minimum-norm required for two sample to be misclassified usually has a larger norm than mask with the mimimum-norm required for each of them.

As for $\text{P}_{\text{MT},i}$, again, Thm. 3.2 from the main paper is also applicable here, but with a slightly different proof (shown below) in adaption to the modifications to the definitions for the patch replacement BPs in Apdx. C.1.

*Proof.* Step 1: Similar to the proof in Apdx. A.2, We show that for any $i \in \mathcal{C}$ and $\mathbf{X}$, $\mathbf{Y}$ i.i.d. following distribution $P_i$, $\mathbf{X} \in \mathcal{T}_\epsilon(\mathbf{Y})$ and $\mathbf{Y} \in \mathcal{T}_\epsilon(\mathbf{X})$ *if and only if* $\mathcal{S}_\epsilon(\mathbf{X}) \cap \mathcal{S}_\epsilon(\mathbf{Y}) \neq \emptyset$.

(a) If $\mathcal{S}_\epsilon(\mathbf{X}) \cap \mathcal{S}_\epsilon(\mathbf{Y}) \neq \emptyset$, there exists $\mathbf{s}$ such that $\mathbf{s} \in \mathcal{S}_\epsilon(\mathbf{X})$ and $\mathbf{s} \in \mathcal{S}_\epsilon(\mathbf{Y})$. By Def. C.1,

$$\mathbf{s} \in \mathcal{S}_\epsilon(\mathbf{X}) \Rightarrow f(\mathbf{\Delta}(\mathbf{X}; \mathbf{m}, \mathbf{u})) \neq f(\mathbf{X}) \tag{49}$$

$$\mathbf{s} \in \mathcal{S}_\epsilon(\mathbf{Y}) \Rightarrow f(\mathbf{\Delta}(\mathbf{Y}; \mathbf{m}, \mathbf{u})) \neq f(\mathbf{Y}) \tag{50}$$

Then, by Definition C.2, the existence of such $\mathbf{s}$ yields $\mathbf{X} \in \mathcal{T}_\epsilon(\mathbf{Y})$ and $\mathbf{Y} \in \mathcal{T}_\epsilon(\mathbf{X})$.

(b) We prove the "only if" part also by contradiction. Given $\mathbf{X} \in \mathcal{T}_\epsilon(\mathbf{Y})$ and $\mathbf{Y} \in \mathcal{T}_\epsilon(\mathbf{X})$, suppose $\mathcal{S}_\epsilon(\mathbf{X}) \cap \mathcal{S}_\epsilon(\mathbf{Y}) = \emptyset$. By Definition C.2, if $\mathbf{Y} \in \mathcal{T}_\epsilon(\mathbf{X})$, there exists $\mathbf{s} = \{\mathbf{m}, \mathbf{u}\} \in \mathcal{S}_\epsilon(\mathbf{X})$ such that $f(\boldsymbol{\Delta}(\mathbf{Y}; \mathbf{m}, \mathbf{u})) \neq f(\mathbf{Y})$. Since $\mathcal{S}_\epsilon(\mathbf{X}) \cap \mathcal{S}_\epsilon(\mathbf{Y}) = \emptyset$, such $\mathbf{s} \notin \mathcal{S}_\epsilon(\mathbf{Y})$; hence, by Definition C.1

$$||\mathbf{m}||_1 - ||\mathbf{m}^*(\mathbf{Y})||_1 > \epsilon. \tag{51}$$

Similarly, there exists $\mathbf{s}' = \{\mathbf{m}', \mathbf{u}'\} \in \mathcal{S}_\epsilon(\mathbf{Y})$ such that $f(\boldsymbol{\Delta}(\mathbf{X}; \mathbf{m}', \mathbf{u}')) \neq f(\mathbf{X})$ and $\mathbf{s}' \notin \mathcal{S}_\epsilon(\mathbf{X})$ (since it is assumed that $\mathcal{S}_\epsilon(\mathbf{X}) \cap \mathcal{S}_\epsilon(\mathbf{Y}) = \emptyset$); hence

$$||\mathbf{m}'||_1 - ||\mathbf{m}^*(\mathbf{X})||_1 > \epsilon. \tag{52}$$

By Definition C.1, for all $\tilde{\mathbf{s}} = \{\tilde{\mathbf{m}}, \tilde{\mathbf{u}}\} \in \mathcal{S}_\epsilon(\mathbf{Y})$, $||\tilde{\mathbf{m}}||_1 - ||\mathbf{m}^*(\mathbf{Y})||_1 < \epsilon$. Thus, we have

$$||\mathbf{m}||_1 - ||\mathbf{m}^*(\mathbf{Y})||_1 > ||\tilde{\mathbf{m}}||_1 - ||\mathbf{m}^*(\mathbf{Y})||_1 \tag{53}$$

and accordingly $||\mathbf{m}||_1 > ||\tilde{\mathbf{m}}||_1$ for all $\tilde{\mathbf{s}} \in \mathcal{S}_\epsilon(\mathbf{Y})$. Similarly, for all $\tilde{\mathbf{s}}' = \{\tilde{\mathbf{m}}', \tilde{\mathbf{u}}'\} \in \mathcal{S}_\epsilon(\mathbf{X})$, we have

$$||\mathbf{m}'||_1 - ||\mathbf{m}^*(\mathbf{X})||_1 > ||\tilde{\mathbf{m}}'||_1 - ||\mathbf{m}^*(\mathbf{X})||_1 \tag{54}$$

and thus, $||\mathbf{m}'||_1 > ||\tilde{\mathbf{m}}'||_1$ for all $\tilde{\mathbf{s}}' \in \mathcal{S}_\epsilon(\mathbf{X})$. Clearly, there is a contradiction since there cannot exist an element in one set having a larger norm than all elements in another set while vise versa; and therefore, $\mathcal{S}_\epsilon(\mathbf{X}) \cap \mathcal{S}_\epsilon(\mathbf{Y}) \neq \emptyset$.

Step 2: We show that the upper bound for $\mathrm{P}_{\mathrm{MT},i}$ goes to 0 when $\epsilon \to 0$.

Note that by Definition C.1, $\mathcal{S}_\epsilon(\mathbf{X}) \cap \mathcal{S}_\epsilon(\mathbf{Y}) \neq \emptyset$ *only if* $\big|||\mathbf{m}^*(\mathbf{X})||_1 - ||\mathbf{m}^*(\mathbf{Y})||_1\big| \leq \epsilon$; also based on Step 1:

$$\mathrm{P}_{\mathrm{MT},i} = \mathrm{P}(\mathcal{S}_\epsilon(\mathbf{X}) \cap \mathcal{S}_\epsilon(\mathbf{Y}) \neq \emptyset) \leq \mathrm{P}\big(\big|||\mathbf{m}^*(\mathbf{X})||_1 - ||\mathbf{m}^*(\mathbf{Y})||_1\big| \leq \epsilon\big). \tag{55}$$

For $\mathbf{X}$ following continuous distribution $P_i$, $\mathbf{m}^*(\mathbf{X})$ also follows some continuous distribution on $(0, +\infty)$. Since $\mathbf{X}$ and $\mathbf{Y}$ are i.i.d., the right hand side of Eq. (55) goes to 0 as $\epsilon \to 0$. $\qquad \square$

Based on the above, we have reached the same conclusions for patch replacement BPs as in the main paper. That is, for the non-attack case, we will likely have $\mathrm{P}_{\mathrm{NT},i} \geq \mathrm{P}_{\mathrm{MT},i}$, and consequently, $\mathrm{ET}_{i,\epsilon} \leq \frac{1}{2}$ based on Thm. 3.1.

**Attack case.** For class $i \in \mathcal{C} = \{0, 1\}$, we consider a successful BA with target class $(1 - i)$. The BP used by the attacker is specified by $\mathbf{s}_0 = \{\mathbf{m}_0, \mathbf{u}_0\}$ with mask $\mathbf{m}_0$ and patch $\mathbf{u}_0$. Thus, for any $\mathbf{X} \sim P_i$, due to the success of the BA, $f(\mathbf{X}) = i$ and $f(\boldsymbol{\Delta}(\mathbf{X}; \mathbf{m}_0, \mathbf{u}_0)) \neq f(\mathbf{X})$. Similar to our discussion in the main paper, $\mathbf{s}_0$ will likely be a common element in both $\mathcal{S}_\epsilon(\mathbf{X})$ and $\mathcal{S}_\epsilon(\mathbf{Y})$ for $\mathbf{X}$, $\mathbf{Y}$ i.i.d. following $P_i$. For the same reason, in this case, the ET statistic will likely be greater than $\frac{1}{2}$.

Similar to additive perturbation BPs, for patch replacement BPs, we also have a guarantee for a large ET statistic ($\mathrm{ET}_{i,\epsilon} = 1$) when the mask size of the BP used by the attacker is sufficiently small. Such property is summarized in the theorem below.

**Theorem C.1.** If class $(1 - i)$ it the target class of a BA with patch replacement BP $\mathbf{s}_0 = \{\mathbf{m}_0, \mathbf{u}_0\}$ such that $||\mathbf{m}_0||_1 \leq \epsilon$, we will have $P(\mathcal{S}_\epsilon(\mathbf{X}) \cap \mathcal{S}_\epsilon(\mathbf{Y}) \neq \emptyset) = 1$ for $\mathbf{X}$ and $\mathbf{Y}$ i.i.d. following $P_i$; and furthermore, $\mathrm{ET}_{i,\epsilon} = 1$.

*Proof.* For any $\mathbf{X} \sim P_i$, since $\mathbf{s}_0 = \{\mathbf{m}_0, \mathbf{u}_0\}$ satisfies $f(\boldsymbol{\Delta}(\mathbf{X}; \mathbf{m}_0, \mathbf{u}_0)) \neq f(\mathbf{X})$, and also because

$$||\mathbf{m}_0||_1 - ||\mathbf{m}^*(\mathbf{X})||_1 < ||\mathbf{m}_0||_1 \leq \epsilon, \tag{56}$$

by Definition C.1, we have $\mathbf{s}_0 \in \mathcal{S}_\epsilon(\mathbf{X})$. Thus, $\mathbf{s}_0 \in \mathcal{S}_\epsilon(\mathbf{X}) \cap \mathcal{S}_\epsilon(\mathbf{Y})$ for $\mathbf{X}$ and $\mathbf{Y}$ i.i.d. following $P_i$; and

$$P(\mathcal{S}_\epsilon(\mathbf{X}) \cap \mathcal{S}_\epsilon(\mathbf{Y}) \neq \emptyset) = 1. \tag{57}$$

The rest of the proof (showing that $\mathrm{ET}_{i,\epsilon} = 1$ for this attack scenario) is exactly the same as the proof of Thm. 3.3 shown in Apdx. A.3, thus is neglected here. $\qquad \square$

Table 3: Details of CIFAR-10, CIFAR-100, STL-10, TinyImageNet, FMNIST, MNIST datasets.

| | Color | Image size | # Classes | # Images/class | # Training images/class |
|---|---|---|---|---|---|
| CIFAR-10 | ✓ | $32 \times 32$ | 10 | 6000 | 5000 |
| CIFAR-100 | ✓ | $32 \times 32$ | 100 | 600 | 500 |
| STL-10 | ✓ | $96 \times 96$ | 10 | 1300 | 500 |
| TinyImageNet | ✓ | $64 \times 64$ | 200 | 600 | 500 |
| FMNIST | ✗ | $28 \times 28$ | 10 | 7000 | 6000 |
| MNIST | ✗ | $28 \times 28$ | 10 | 7000 | 6000 |

## C.3 DETECTION PROCEDURE FOR PATCH REPLACEMENT BP

The same procedure, i.e. Alg. 1 in the main paper, can be used for detecting BAs with patch replacement BP, with only the following two modifications. We only need to first replace line 8 of Alg. 1 by:

"Obtain an empirical solution $\hat{s}(x_n^{(i)}) = \{\hat{m}(x_n^{(i)}), \hat{u}(x_n^{(i)})\}$ to problem (46) using random initialization."

and then change line 9 of Alg. 1 to:

"$\widehat{\mathcal{T}}_\epsilon(x_n^{(i)}) \leftarrow \widehat{\mathcal{T}}_\epsilon(x_n^{(i)}) \cup \{x_k^{(i)} | k \in \{1, \cdots, N_i\} \setminus n, f(\Delta(x_k^{(i)}; \hat{m}(x_n^{(i)}), \hat{u}(x_n^{(i)}))) \neq f(x_k^{(i)})\}$".

Such a simple "module-based" modification allows our detection framework to be applicable to a variety of BP embedding mechanisms, which again, shows the generalization capability of our detection framework.

# D DETAILS OF EXPERIMENT SETTINGS

## D.1 DETAILS OF DATASETS

Our experiments are conducted on six popular benchmark image datasets. They are CIFAR-10, CIFAR-100 Krizhevsky (2012), STL-10 Coates et al. (2011), TinyImageNet, FMNIST Xiao et al. (2017) and MNIST Lecun et al. (1998). All the datasets are associated with the *torchvision* package, except for that STL-10 is downloaded from the official website `https://cs.stanford.edu/~acoates/stl10/`. Though the details of these datasets can be easily found online, we summarize them in Tab. 3.

## D.2 DETAILS FOR GENERATING THE 2-CLASS DOMAINS

In Sec. 4.1, we generate 45 2-class domains from CIFAR-10, and 20 2-class domains from each of CIFAR-100, STL-10, TinyImageNet, FMNIST, and MNIST. Here we provide more details about how these 2-class domains are generated.

As mentioned in Sec. 4.1, for CIFAR-10, the 45 2-class domains are corresponding to the 45 unordered class pairs of CIFAR-10 respectively. For each of CIFAR-100, FMNIST, and MNIST, we randomly sample 20 unordered class pairs, each forming a 2-class domain. For TinyImageNet, due to high image resolution and data scarcity, we generate 20 "super class" pairs – for each pair, we randomly sample 20 classes from the original category space and then evenly assign them to the two super classes (each getting 10 classes from the original category space). Similarly, for STL-10 with 10 classes, we generate 20 super class pairs by randomly and evenly dividing the 10 classes into two groups (of 5 classes from the original category space) for each pair. For each generated 2-class domain, we use the subset of data associated with these two (super) classes from the original dataset, with the original train-test split.

## D.3 DETAILS OF BPs

In this paper, we consider both additive perturbation BPs embedded by Eq. (1) and patch replacement BPs embedded by Eq. (2) that are frequently used in existing backdoor papers. Despite the BPs (with images embedded with them) illustrated in Fig. 1 in the main paper, here, in Fig. 5, we show

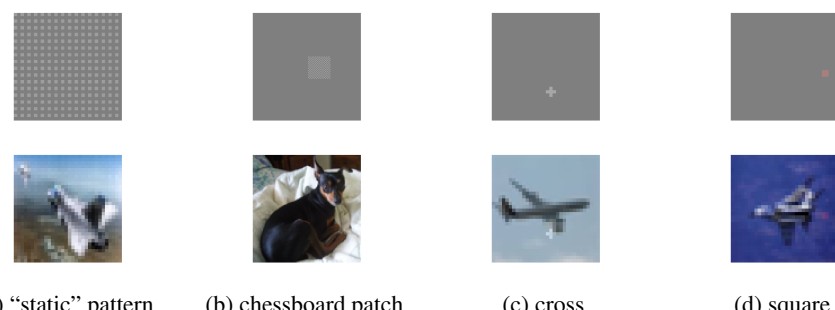

| (a) "static" pattern | (b) chessboard patch | (c) cross | (d) square |

Figure 5: BPs used in our experiments that are not shown in Fig. 1 of the main paper due to space limitations; and images with these BPs embedded. BPs in (a) and (b) are amplified for visualization.

the BPs used in our experiments that are not included in the main paper due to space limitations. In the following, we also provide details for each of these BPs (in both Fig. 1 and Fig. 5).

First, we provide the details for all the additive perturbation BPs. The "chessboard" pattern in Fig. 1a is a "global" pattern that has been used by Xiang et al. (2020). Here, one and only one of two adjacent pixels are perturbed positively by 3/255 in all color channels. Another global pattern is the "static" pattern in Fig. 5a considered by both Zhong et al. (2020); Xiang et al. (2021b). For pixel indices starting from 0, a pixel $(i, j)$ is perturbed positively if and only if $i$ and $j$ are both even numbers. Again, the perturbation size is 3/255 for all pixels being perturbed.

Other additive perturbation BPs are all "localized" patterns. The "L" pattern in Fig. 1b and the "X" pattern in Fig. 1c have been used by both Tran et al. (2018) and Wang et al. (2020). For the "L" pattern, we perturb all the color channels by 50/255. For the "X" pattern, for each attack, we randomly choose a channel (for all images to be embedded in for this particular attack) and perturb the associated pixels positively by 50/255. The "pixel" BP in Fig. 1d has been used by Tran et al. (2018); Chen et al. (2018), where a single pixel is perturbed in all channels by 50/255 for color images and 70/255 for gray-scale images. The "chessboard patch" pattern in Fig. 5b, the "cross" in Fig. 5c, and the "square" in Fig. 5d have all been previously considered. For the cross and the chessboard patch, the perturbation sizes for each pixel being perturbed are 50/255 and 5/255, respectively; and perturbation is applied to all channels. For the square pattern, one channel is randomly selected for each attack and the perturbation size is 50/255. The spatial location of all these localized patterns are randomly selected over the entire image (and fixed for all images to be embedded in) for each attack. Only for gray-scale images, the pixels being perturbed are restricted to one of the four corners, such that these pixels will likely be black (with pixel value close to 0) originally.

Next, we provide the details for the two patch replacement BPs considered in our experiments. The BP in Fig. 1e is a small, monochromatic patch located near the margin of the images to be embedded in. Similar BPs have been considered by Gu et al. (2019) and Wang et al. (2019). The color is randomly chosen and fixed for each attack. The BP in Fig. 1f is a small noisy patch located near the margin of the images to be embedded in. Similar BPs have been considered by Turner et al. (2019) and A. Saha (2020). For both BPs, once the location is selected, the same location will be applied to all images to be embedded in for the same attack. Also, for both BPs, the size of the patch is $3 \times 3$ for the 2-class domains generated from CIFAR-10 and CIFAR-100; $4 \times 4$ for the 2-class domains generated from TinyImageNet; and $10 \times 10$ for the 2-class domains generated from STL-10.

## D.4 OTHER ATTACK CONFIGURATIONS

In the main paper, we defined a "code" for each ensemble of attack instances based on both the type of BP being used and the dataset from which the associated 2-class domains are generated. Here, we summarize these codes in Tab. 4 for reference. Also in the main paper, we have described most of the attack configurations for the attacks instances in each ensemble. For some ensembles, each attack instance is associated with two attacks, each with one of the two class being the target class. Also, the BPs used by the two attacks, though of the same type, are guaranteed to be sufficiently different in shapes (for additive perturbation BP) or colors (for patch replacement BP). For other

Table 4: Short hand "code" for each ensemble of attack instances based on both the BP being used and the dataset where the associated 2-class domains are generated from. "n/a" represents "not applicable".

|  | Additive perturbation BP | Patch replacement BP |
|---|---|---|
| CIFAR-10 | $A_1$ | $A_7$ |
| CIFAR-100 | $A_2$ | $A_8$ |
| STL-10 | $A_3$ | $A_9$ |
| TinyImageNet | $A_4$ | $A_{10}$ |
| FMNIST | $A_5$ | n/a |
| MNIST | $A_6$ | n/a |

Table 5: Summary of attack configurations for instances in each of ensembles $A_1$-$A_{10}$. For each ensemble, we show the number of attacks for each instance in this ensemble. We also show, for each ensemble, the number of samples (embedded with BP and labeled to the target class) used for poison the training set for each attack associated with this ensemble, as well as the corresponding poisoning rate. Poisoning rate is defined as the number of samples inserted into the training set by the attacker divided by the total number of training samples from the target class after poisoning.

|  | $A_1$ | $A_2$ | $A_3$ | $A_4$ | $A_5$ | $A_6$ | $A_7$ | $A_8$ | $A_9$ | $A_{10}$ |
|---|---|---|---|---|---|---|---|---|---|---|
| # Attacks | 2 | 2 | 2 | 1 | 1 | 1 | 2 | 2 | 1 | 2 |
| # Poison samples | 500 | 150 | 1000 | 1500 | 1000 | 1000 | 500 | 50 | 750 | 500 |
| Poisoning rate | 9.1% | 23.0% | 28.6% | 23.0% | 14.3% | 14.3% | 9.1% | 9.1% | 23.1% | 9.1% |

ensembles, there is only one attack for each attack instance. Here, we provide a summary for these configurations in Tab. 5. Also, in Tab 5, for attack instances in each ensemble, we summarize the number of training samples embedded with the BP and labeled to the target class that are used for poisoning the classifier's training set for the associated attacks.

### D.5 TRAINING DETAILS

Here we provide the training details that are not included in the main paper due to space limitations. For each generated 2-class domain, we use the same training configuration irrespective of the existence of BA. In Tab. 6, we show the training details including learning rate, batch size, number of epochs, whether or not using training data augmentation, choice of optimizer (Adam D. P. Kingma (2015) or stochastic gradient descent (SGD)) for 2-class domains generated from CIFAR-10, CIFAR-100, STL-10, TinyImageNet, FMNIST, and MNIST, respectively. Training data augmentations for 2-class domains generated from TinyImageNet include random cropping and random horizontal flipping – these augmentations are helpful for the backdoor mapping to be learned without compromising the classifier's accuracy on clean test samples. Otherwise, we may not easily produce an effective attack to evaluate the performance of our defense.

We also show the effectiveness of the attacks we created for evaluating our defense. Commonly, the effectiveness of a BA is evaluated by *attack success rate* (ASR) and *clean test accuracy* (ACC) Xiang et al. (2020); Wang et al. (2020). ASR is defined (for each attack) as the probability that a test image from the source class is (mis)classified to the target class of BA when the BP is embedded. ACC is defined (for each classifier being attack regardless of the number of attacks) as the classification accuracy on test samples with no BP. In our experiments, we evaluate ASR and ACC using images from the test set associated with each 2-class domain – these images are not used during training. For each attack instance, we evaluate ASR for each attack (since there can be either one attack or two attacks with different BA target class) separately. In Tab. 7, for each of ensembles $A_1$-$A_{10}$, we show the average ACC for the classifier being attacked over all instances in the ensemble; we also show the mean and minimum ASR over all attacks of all instances in the ensemble. As a reference,

Table 6: Training details, including learning rate, batch size, number of epochs, whether or not using training data augmentation, choice of optimizer (Adam D. P. Kingma (2015) or stochastic gradient descent (SGD)), for 2-class domains generated from CIFAR-10, CIFAR-100, STL-10, TinyImageNet, FMNIST, and MNIST, respectively.

| | Learning rate | Batch size | # Epochs | Data augmentation | Optimizer |
|---|---|---|---|---|---|
| CIFAR-10 | 0.001 | 32 | 150 | ✗ | Adam |
| CIFAR-100 | 0.001 | 32 | 150 | ✗ | Adam |
| STL-10 | 0.001 | 128 | 150 | ✗ | Adam |
| TinyImageNet | 0.001 | 128 | 150 | ✓ | Adam |
| FMNIST | 0.01 | 256 | 100 | ✗ | SGD |
| MNIST | 0.01 | 256 | 100 | ✗ | SGD |

Table 7: Average clean test accuracy (ACC, in percentage) over all classifiers being attacked, average and minimum attack success rate (ASR, in percentage) over all attacks, for each of ensemble $A_1$-$A_{10}$ of attack instances.

| | $A_1$ | $A_2$ | $A_3$ | $A_4$ | $A_5$ | $A_6$ | $A_7$ | $A_8$ | $A_9$ | $A_{10}$ |
|---|---|---|---|---|---|---|---|---|---|---|
| ASR (average) | 95.7±3.6 | 91.6±4.5 | 98.8±1.0 | 93.9±2.8 | 96.8±3.3 | 99.8±0.4 | 99.2±1.1 | 96.4±4.0 | 97.9±1.4 | 94.5±4.6 |
| ASR (minimum) | 82.3 | 80.0 | 95.9 | 88.0 | 87.6 | 98.4 | 92.5 | 82.0 | 95.1 | 83.2 |
| ACC (average) | 94.6±4.2 | 90.7±4.5 | 79.6±2.9 | 77.0±3.2 | 99.0±1.1 | 99.8±0.1 | 96.7±2.5 | 93.2±4.1 | 78.7±3.4 | 77.2±3.5 |

in Tab. 8 for each of ensemble $C_1$-$C_6$ of clean instances, we show the average ACC over all the clean classifiers for the ensemble. Based on the results in both Tab. 7 and Tab. 8, all the attacks we created are successful with high ASR and almost no degradation in ACC.

### D.6 BP Reverse-Engineering Algorithms

In the main paper, we evaluate our detection framework with two BP reverse-engineering algorithms, which are denoted as RE-AP and RE-PR, respectively. RE-AP is proposed by Xiang et al. (2020) for reverse-engineering additive perturbation BPs. The general form of RE-AP estimates a common perturbation that induces a group of images to be misclassified to a common target class. When there is a single image in such a group, and when there are only two classes, the optimization problem solved by RE-AP is reduced to (3). To solve this problem for some target class $i \in \mathcal{C}$ and image $\mathbf{x} \in \mathcal{X}$ from the class other than $i$, RE-AP minimizes the following surrogate objective function:

$$L_{\text{AP}}(\mathbf{v}) = -\log p(i|\mathbf{x} + \mathbf{v}), \tag{58}$$

using gradient descent with $\mathbf{v}$ initialized from $\mathbf{0}$, until the constraint of (3) is satisfied. Here, $p(i|\mathbf{x})$ denotes the classifier's posterior of class $i$ give any input sample $\mathbf{x} \in \mathcal{X}$. The step size for minimization is set small to ensure a good "quality" for the solution; otherwise, the resulting perturbation may have a much larger norm than the minimum norm perturbation required for inducing a misclassification. Moreover, for each domain and each classifier to be inspected, choosing a proper step size can be done based on the norm of the solution and without any knowledge of the presence of BA.

Another BP reverse-engineering algorithm, RE-PR is proposed by Wang et al. (2019) for patch replacement BPs. Similarly, RE-PR solves problem (46) in Apdx. C.1, which is the counterpart of problem (3) for patch replacement BPs. Formally, for some target class $i \in \mathcal{C}$ and image $\mathbf{x} \in \mathcal{X}$ from the class other than $i$, RE-PR minimizes the following surrogate objective function:

$$L_{\text{PR}}(\mathbf{m}, \mathbf{u}) = -\log p(i|\mathbf{\Delta}(\mathbf{x}; \mathbf{m}, \mathbf{u})) + \lambda||\mathbf{m}||_1, \tag{59}$$

using gradient descent, where $\lambda$ is the Lagrange multiplier and $\mathbf{\Delta}(\mathbf{x}; \mathbf{m}, \mathbf{u})$ is the alternative expression to Eq. 2 (see the description below Eq. (46)) for patch replacement BP embedding.

Table 8: Average clean test accuracy (ACC, in percentage) over the classifiers for the clean instances in each of ensemble $C_1$-$C_6$.

| | $C_1$ | $C_2$ | $C_3$ | $C_4$ | $C_5$ | $C_6$ |
|---|---|---|---|---|---|---|
| ACC (average) | 95.4±3.6 | 93.3±3.3 | 80.5±3.2 | 78.2±3.0 | 99.3±1.0 | 99.8±0.1 |

### D.7 LIMITATIONS OF THE COSINE SIMILARITY STATISTIC IN BA DETECTION

In the main paper, we compared our ET statistic with other three types of detection statistics including a cosine similarity (CS) statistic proposed by Wang et al. (2020). As an important work addressing unsupervised backdoor detection without access to the training set, this method can effectively detect backdoor attacks when their are multiple classes with only few of them are backdoor target classes.

For general classification domains with arbitrary number of classes, the CS statistic is obtained for each putative target class $t \in \mathcal{C}$ as following. First, a *common* (patch replacement) BP is estimated to: a) induces a group of images from classes other than $t$ to be misclassified in an untargeted fashion (i.e., to any class other than their originally labeled classes); b) not induce any class $t$ images to be misclassified; and c) have as small mask size (measured by $l_1$ norm) as possible. Accordingly, Wang et al. (2020) proposed to minimize the following loss:

$$
\begin{aligned}
L_{\text{CSC}}(\mathbf{m}, \mathbf{u}) = & \sum_{i \in \mathcal{C} \backslash t} \sum_{\mathbf{x} \in \mathcal{D}_i} \max\{h_i(\boldsymbol{\Delta}(\mathbf{x}; \mathbf{m}, \mathbf{u})) - \max_{j \neq i} h_j(\boldsymbol{\Delta}(\mathbf{x}; \mathbf{m}, \mathbf{u})), -\kappa\} \\
& + \sum_{\mathbf{x} \in \mathcal{D}_t} \max\{\max_{j \neq t} h_j(\boldsymbol{\Delta}(\mathbf{x}; \mathbf{m}, \mathbf{u})) - h_t(\boldsymbol{\Delta}(\mathbf{x}; \mathbf{m}, \mathbf{u})), -\kappa\} + \lambda ||\mathbf{m}||_1,
\end{aligned}
\tag{60}
$$

where $h_i(\cdot) : \mathcal{X} \to \mathbb{R}$ is the logit (right before softmax) of class $i \in \mathcal{C}$ Carlini & Wagner (2017). We denote the estimated (common) BP for class $t$ as $\mathbf{s}_t^* = \{\mathbf{m}_t^*, \mathbf{u}_t^*\}$.

Then, for each image not from class $t$, a sample-wise BP is estimated to: a) induce the image to be misclassified to class $t$; and b) have as small mask size (measured by $l_1$ norm) as possible. Thus, the following loss is minimized for each $\mathbf{x} \in \cup_{i \in \mathcal{C} \backslash t} \mathcal{D}_i$:

$$
L_{\text{CSS}}(\mathbf{m}, \mathbf{u}) = \max\{\max_{j \neq t} h_j(\boldsymbol{\Delta}(\mathbf{x}; \mathbf{m}, \mathbf{u})) - h_t(\boldsymbol{\Delta}(\mathbf{x}; \mathbf{m}, \mathbf{u})), -\kappa\} + \lambda ||\mathbf{m}||_1.
\tag{61}
$$

We denote the sample-wise BP estimated for class $t$ and sample $\mathbf{x}$ as $\tilde{\mathbf{s}}_t^*(\mathbf{x}) = \{\tilde{\mathbf{m}}_t^*(\mathbf{x}), \tilde{\mathbf{u}}_t^*(\mathbf{x})\}$.

Finally, the cosine similarity statistic for class $t$ is computed by:

$$
\text{CS}_t = \frac{1}{|\cup_{i \in \mathcal{C} \backslash t} \mathcal{D}_i|} \sum_{\mathbf{x} \in \cup_{i \in \mathcal{C} \backslash t} \mathcal{D}_i} \cos(\mathbf{z}(\boldsymbol{\Delta}(\mathbf{x}; \mathbf{m}_t^*, \mathbf{u}_t^*)), \mathbf{z}(\boldsymbol{\Delta}(\mathbf{x}; \tilde{\mathbf{m}}_t^*(\mathbf{x}), \tilde{\mathbf{u}}_t^*(\mathbf{x})))),
\tag{62}
$$

where $\mathbf{z}(\cdot) : \mathcal{X} \to \mathbb{R}^d$ is the mapping from input layer to the penultimate layer with some dimension $d$. $\cos(\cdot) : \mathbb{R}^d \times \mathbb{R}^d \to [-1, 1]$ is the cosine similarity between two real vectors.

Based on our results in Fig. 2, CS for BA target classes and non-target classes are separable for most domains. This is not surprising because when class $t \in \mathcal{C}$ is a BA target class, the estimated common BP will likely be highly correlated with the sample-wise BPs estimated for images from classes other than $t$; thus, the resulting CS will be large and possibly close to 1. If class $t \in \mathcal{C}$ is not a BA target class, the estimated common BP may induce images from classes other than $t$ to be misclassified to some arbitrary classes (possibly the "semantically" closest class for each individual image). Thus, the common BP may be very different from sample-wise BP estimated to only induce misclassifications to class $t$. Consequently, the CS statistic will likely be small.

However, considering our 2-class problem with no sufficient number of statistics to inform estimation of a null distribution, and also our assumption that there is no domain-specific supervision (e.g. using clean classifiers trained for the same domain) for setting a proper detection threshold, CS statistic may not be effective since it is sensitive to domains and DNN architectures.

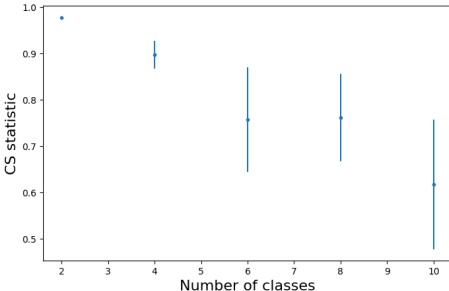

Figure 6: Average CS statistic versus the number of classes in the domain.

First, when there is no BA, for ReLU networks, the penultimate layer features are always non-negative, such that the cosine similarity is guaranteed to be non-negative. However, for DNN with sigmoid or leaky ReLU activation functions, the penultimate layer features can be negative. Thus, the CS statistic may be distributed in the entire interval of $[-1, 1]$. Such large difference in the null distribution of CS statistic makes the choice of a detection threshold very difficult without domain-specific knowledge.

Second, CS is also sensitive to the classification domain, especially the number of classes. Considering some putative target class $t \in \mathcal{C}$, when there are a large number of classes in the domain, the common BP estimated for a group of images from classes other than $t$ will likely be very different with the sample-wise BP estimated for each individual in the group. In particular, most images will likely be misclassified to some class other than $t$ when the common BP is embedded; but the sample-wise BP is estimated for each of these images to induce them to be misclassified to class $t$. Thus, the penultimate layer feature associated with the common BP and the sample-wise BP will likely be different for most images. However, when there are only two classes, i.e. $\mathcal{C} = \{0, 1\}$ in our case, the images used for BP estimation for class $t$ are all from class $(1 - t)$. Moreover, both common BP and samples-wise BP estimated for these images will induce them to be misclassified to class $t$. Thus, CS obtained for this case will likely be larger than for the cases where there are a large number of classes when there is no BA. We demonstrate this phenomenon in the following.

We construct five domains from CIFAR-10. The first four domains contain 2, 4, 6, 8 classes randomly selected from the 10 classes of CIFAR-10 respectively, and the the fifth domain is the original CIFAR-10 with 10 classes. We train a classifier without BA for each domain using the same configurations as in Sec. 4.1. For each classifier, we obtain CS statistics for all classes. In Fig. 6, we show the average CS statistic over all classes for the five classifiers. In general, CS statistic decreases as the number of classes grows; thus, it is highly domain-dependent.

### D.8    DETAILS FOR MULTI-CLASS EXPERIMENTS

In Sec. 4.4, we evaluate the performance of our detection framework for multi-class scenarios with arbitrary number of attacks. In other words, the classifier has more than two classes, and each class can possibly be a BA target class.

For each of CIFAR-10, CIFAR-100, and STL-10, we create three attack instances with one, two, and three attacks, respectively. Like the attacks we created for the 2-class domains, the attacks here are created following the same data poisoning protocol that has been widely considered in existing works. That is, we create backdoor training images by embedding a BP into a small set of images from classes other than the target class. These backdoor training images are labeled to a target class and inserted into the training set of the classifier Gu et al. (2019). In our experiment here, for each attack of each instance, we randomly select a target class. For simplicity, we consider only additive perturbation BP here. We randomly select a shape (and location for localized BP) for the BP to be used from our pool of candidate BPs. Note that for any two attacks of the same attack instance, the target classes and the BPs should both be different from each other. For attacks on CIAFR-10, CIFAR-100, and STL-10, the backdoor training images are created using 60, 10, and 100 clean images per class (not including the target class), respectively.

Table 9: Attack success rate (ASR) and clean test accuracy (ACC) for the classifiers being attacked (with one, two, and three attacks/target classes) for CIAFR-10, CIFAR-100, and STL-10, respectively; and ACC for the clean classifiers trained for the three domains respectively.

| | CIFAR-10 | CIFAR-100 | STL-10 |
|---|---|---|---|
| one attack | ASR: 97.0
ACC: 93.7 | ASR: 97.7
ACC: 71.5 | ASR: 95.1
ACC: 79.7 |
| two attacks | ASR=96.0, 93.7
ACC: 92.5 | ASR=89.6, 95.3
ACC: 71.9 | ASR=99.6, 96.0
ACC: 80.8 |
| three attacks | ASR: 94.5, 80.2, 97.9
ACC: 92.8 | ASR: 95.7, 74.7, 97.4
ACC: 70.5 | ASR: 78.1, 99.0, 95.1
ACC: 79.1 |
| no attack | ACC: 92.5 | ACC: 70.4 | ACC: 78.8 |

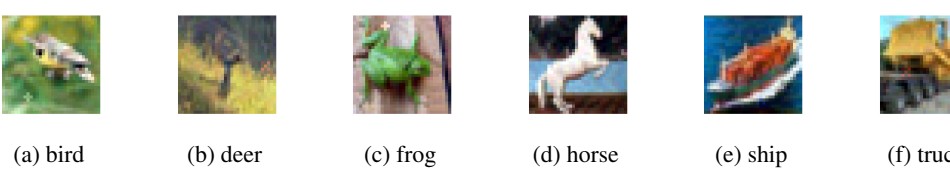

| (a) bird | (b) deer | (c) frog | (d) horse | (e) ship | (f) truck |
|---|---|---|---|---|---|

Figure 7: Examples for backdoor training images for clean-label BAs. These images are originally from the BA target class, perturbed (in human-imperceptible fashion) to be misclassified by a surrogate classifier, embedded with the BP, and are still labeled to the target class.

For each domain, the three attack instances and the clean classifier use the same training configuration. For CIFAR-10 and STL-10, we use ResNet-18 as the DNN architecture; for CIFAR-100, we use ResNet-34 architecture. For all three domains, training data augmentations including random cropping and random horizontal flipping are adopted. Other configurations for classifier training for these domains are the same as for the 2-class domains generated from these original domains (shown in Tab. 6). Using these training configurations, the resulting nine classifiers being attacked (three classifiers for the three attack instances respectively for each domain) all have high attack success rate (ASR). Compared with the three clean classifiers (without BA) trained for the three domains respectively, there is also no significant degradation in clean test accuracy (ACC) for the classifiers being attacked. ASR and ACC for these classifiers are shown in Tab. 9.

# E    USING ET TO DETECT CLEAN-LABEL BAs

In this section, we demonstrate the effectiveness of our detection framework against a recent clean-label BA proposed by Turner et al. (2019). Clean-label BAs are motivated by the possible human/machine inspection of the training set. For example, backdoor training samples labeled to some target class inserted by a typical backdoor attacker are originally from classes other than the target class. Such "mislabeling" may be noticed by a human expert who inspects the training set manually, or may be detected by a shallow neural network trained on a small held-out validation set that is guaranteed to be clean. Thus, Turner et al. (2019) proposed to create backdoor training samples (that will be inserted into the classifier's training set) by embedding the BP only to target class samples. However, for target class samples embedded with the BP, there is no guarantee that it is the BP but not the features associated with the target class that will be learned by the classifier. Thus, Turner et al. (2019) proposed to "destroy" these target class features before embedding the BP when creating backdoor training samples. Then, the classifier will learn the BP and classify any test sample embedded with the BP to the target class.

One simple yet effective approach proposed by Turner et al. (2019) to destroy the features in the backdoor training samples associated with the target class is inspired by a method for creating adversarial examples. The backdoor attacker needs to first train a surrogate classifier using an independently collected clean dataset. Then, for each of a small set of samples from the target class used for creating backdoor training samples, the attacker independently launches a projected gradi-

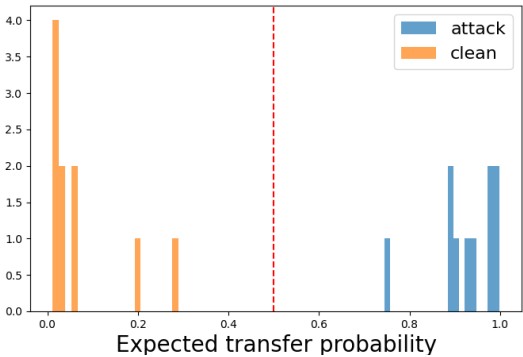

Figure 8: Effectiveness of our defense, with the constant ET threshold 1/2, against the clean-label BA proposed by Turner et al. (2019). Classifiers being attacked have a maximum ET (over the two classes) greater than 1/2; clean classifiers have a maximum ET (over the two classes) less than 1/2.

ent descent (PGD) attack (Madry et al. (2018)) to have the sample be predicted to any class other than its original class (i.e. the target class) by the surrogate classifier. These samples with the target class features destroyed are then embedded with the BP, are still labeled to the target class, and are inserted into the classifier's training set.

In our experiment here, we randomly generate ten 2-class domains from CIFAR-10 following its original train-test split. For each 2-class domain, we first train a surrogate classifier using a subset of the training set (2000 training images per class). The remaining samples (3000 per class) are assumed to be possessed by the trainer for training the victim classifier. For each 2-class domain, we create one attack instance with one BA targeting on the second class and using an additive perturbation BP. The candidate BPs to be used are the same as in our experiments in the main paper. Here, for each BA, we use 1500 images from the target class (i.e. the second class of the associated 2-class domain) to create backdoor training images. These images are randomly sampled from the images used for training the surrogate classifier. For each of these 1500 images, we independently generate an adversarial perturbation using the surrogate classifier following the standard protocol of PGD Madry et al. (2018). In particular, we set the maximum perturbation size as 8/255, the number of perturbation steps as 10, and the step size as 1/255. Most of the perturbed images are misclassified by the surrogate classifier. Then we embed the BP randomly selected from the candidates for each attack into these images with the adversarial perturbation and still label them to the target class, where they are originally from. The created backdoor training images are inserted into the training set of the victim classifier. They will be barely noticeable to human inspectors since they visually look like standard target class images and the embedded BP is almost imperceptible by humans. Some examples of backdoor training images are shown in Fig. 7.

For each attack instance, we use the same training configurations as in Sec. 4.1 of the main paper to train the victim classifier. We also train a clean classifier for each instance to evaluate false detections. Moreover, we apply our detection framework with the BP reverse-engineering algorithm RE-AP and the same defense configurations as in the main paper to both the classifiers being attacked and the clean classifiers. In Fig. 8, we show the maximum ET (over the two classes) for all these classifiers. Using ET and the constant detection threshold 1/2, we perfectly detect all clean-label BAs with no false detections.

# F   EXPERIMENTAL VERIFICATION OF PROPERTY 3.1 FOR PATCH REPLACEMENT BPS

We verify Property 3.1 for patch replacement BPs embedded by Eq. (2). Similar to Sec. 4.3 of the main paper, here, we show that when class $(1 - i)$ for $i \in \mathcal{C} = \{0, 1\}$ is not a BA target class, for any two samples from class $i$, the minimum $l_1$ norm for any common mask that induces both samples to be misclassified will likely be larger than the minimum $l_1$ norm for the masks inducing each individual sample to be misclassified. The masks are obtained by solving problem (3).

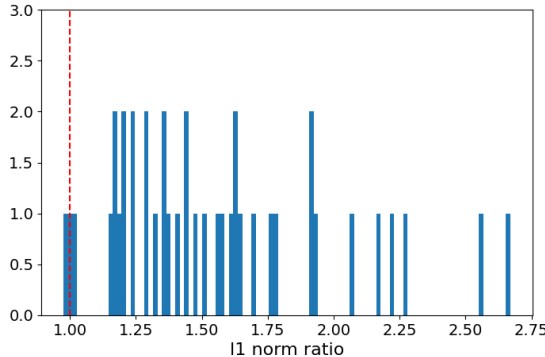

Figure 9: Histogram of $l_1$ norm ratio between pair-wise common mask and maximum of the two sample-wise masks for each random image pair for clean classifiers.

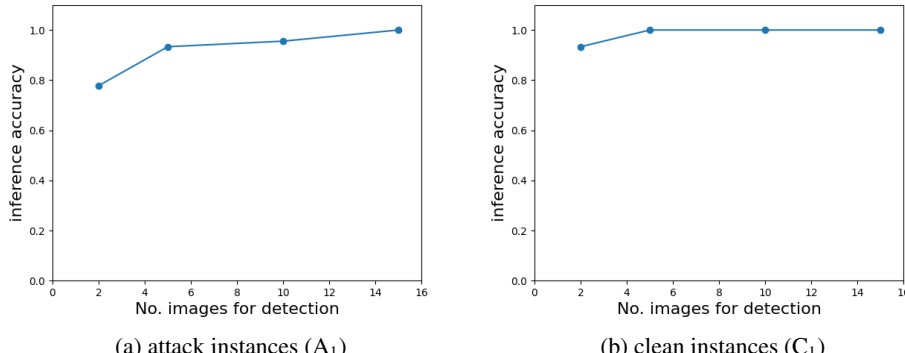

(a) attack instances (A$_1$)            (b) clean instances (C$_1$)

Figure 10: Accuracy of detection inference on the ensemble of attack instances $A_1$ and the ensemble of clean instances $C_1$, when the number of images used for detection varies in [2, 5, 10, 15].

We randomly choose one clean classifier from each of $C_1$-$C_4$. For each classifier, we randomly choose 10 pairs of clean images from a random class of the associated 2-class domain. For each pair of images, we apply the RE-PR algorithm (for reverse-engineering patch replacement BPs) on the two images jointly (to get a pair-wise common mask) as well as separately (to get two sample-wise masks for the two images respectively). Again, we divide the $l_1$ norm of the pair-wise (common) mask by the maximum $l_1$ norm of the two sample-wise masks to get a ratio. In Fig. 9, we plot the histogram of such ratio for all image pairs for all four classifiers – the ratio for most of the image pairs is greater than 1 (marked by the red dashed line in Fig. 9). For these pairs, it is very likely that the BP (in particular, the mask) estimated for one sample cannot induce the other sample to be misclassified and vise versa. Note that if for any image pair, the mask (and some associated pattern) estimated for one sample can induce the other to be misclassified, such mask (and the associated pattern) will be a common mask (and pattern) that induces both images to be misclassified. Then, we would expect the ratio computed above to be very close to 1 for such an image pair.

## G  INFLUENCE OF THE NUMBER OF CLEAN IMAGES FOR DETECTION

The core of our detector is to estimate the ET statistic for each class. Note that the ET statistic is in fact an expectation. In principle, with fewer clean images per class for ET estimation, the variance of the estimated ET will be larger, though the execution time for ET estimation may be smaller. Thus, we would expect that, sometimes, the ET estimated using only a few clean images may be smaller than $\frac{1}{2}$ for the attack case; or larger than $\frac{1}{2}$ for the clean case.

In Sec. 4.1, we used 20 images per class (40 images in total) for backdoor detection for all attack instances and all clean instances, and achieved good detection accuracy. Here, we show the influ-

ence of the number of clean images per class on detection accuracy. In particular, for all 45 attack instances and all 45 clean instances of two-class domains generated from CIFAR-10 (i.e. $A_1$ and $C_1$), we apply the same detector in Sec. 4.1 with detection threshold $\frac{1}{2}$, but varying the number of clean images per class (in [2, 5, 10, 15]) used for detection (i.e. ET estimation). As shown in Fig. 10, with 5 clean images per class (10 images in total), our method achieves relatively good detection accuracy. Even with only 2 clean images per class (which is the minimum sample size for empirical ET estimation), our detector catches $\sim 80\%$ of attacks with less than 10% false detection rate[5].

## H    CHOICE OF THE PATIENCE PARAMETER $\tau$

In all experiments in the main paper, we set the patience parameter in Alg. 1 to $\tau = 4$ and claim that larger $\tau$ will not induce much change to the estimated ET. Here, we provide some empirical evidence to support this claim.

We apply Alg. 1 with RE-AP to a classifier being attacked in $A_1$ and a classifier being attacked in $A_2$. We also apply Alg. 1 with RE-PR to a classifier being attacked in $A_7$ and a classifier being attacked in $A_8$. Moreover, Alg. 1 with both RE-AP and RE-PR are applied to a clean classifier in $C_1$ and a clean classifier in $C_2$. Note that classification domains associated with $A_1$, $A_7$, and $C_1$ are generated from CIFAR-10; while classification domains associated with $A_2$, $A_8$, and $C_2$ are generated from CIFAR-100.

For all experiments in this section, we set the patience parameter to $\tau = 8$ instead of $\tau = 4$ used in the main paper. The purpose is to get a better observation of the asymptotic behavior of $p_n^{(i)} = |\widehat{\mathcal{T}}_\epsilon(\mathbf{x}_n^{(i)})|/(N_i - 1)$ during ET estimation for each clean sample $\mathbf{x}_n^{(i)}$ used for detection (see line 7-12 of Alg. 1 for the definition of related quantities). The number of clean samples used for detection is 20.

As shown in Fig. 11, when applying our method to classifiers being attacked, $p_n^{(i)}$ (for some class $i$) quickly grows to 1 (in very few iterations) for most clean samples used for detection (see (a)(b)(e)(f) of Fig. 11). For a few clean samples, $p_n^{(i)}$ quickly grows to a large value close to 1 and then slowly reaches 1 (see (e)(f) of Fig. 11). Only for very few samples, $p_n^{(i)}$ stays at some value in between 0 and 1 (see (b)(e) of Fig. 11). Based on these observations, which are generally true for other domains we investigated, $\tau = 4$ is not a critical choices to our detection performance. The estimated ET, which is the average $p_n^{(i)}$ for all clean samples used for detection, will likely be greater than $\frac{1}{2}$, as determined by the majority of clean samples used for detection.

On the other hand, when applying our method to clean classifiers, with RE-AP, $p_n^{(i)}$ stays at 0 (or some small values close to 0) for all clean samples used for detection (see (c)(g) of Fig. 11). When applying our method with RE-PR to the same classifiers, $p_n^{(i)}$ stays at 0 or some small values close to 0 for most samples and shows a trend of convergence (see (d)(h) of Fig. 11). Again, reducing $\tau$ from 8 to 4 or further increasing $\tau$ will not change the estimated ET much – the estimated ET for these clean instances will still be clearly less than $\frac{1}{2}$.

## I    USING SYNTHESIZED IMAGES FOR BACKDOOR DETECTION

For most REDs, the defender is assumed to possess a small, clean dataset (collected independently) for detection Wang et al. (2019); Xiang et al. (2020); Guo et al. (2019); Wang et al. (2020). Although this assumption is relatively mild and feasible in most practical scenarios, it may be unnecessary if the defender is able to synthesize the images used for detection.

The first trial was made by Chen et al. (2019), where on simple datasets like MNIST Lecun et al. (1998), images for backdoor pattern reverse-engineering are synthesized by model inversion Fredrikson et al. (2015). However, images generated in such a way are not guaranteed to be visually typical to their designated classes, especially for complicated domains like ImageNet. Thus we ask the following question:

---

[5]Here, we claim a failure in detection if ET is exactly $\frac{1}{2}$ for both clean instances and attack instances. Thus, the actual detection accuracy should be higher than those in Fig. 10 if we either do or do not trigger an alarm when ET is exactly $\frac{1}{2}$.

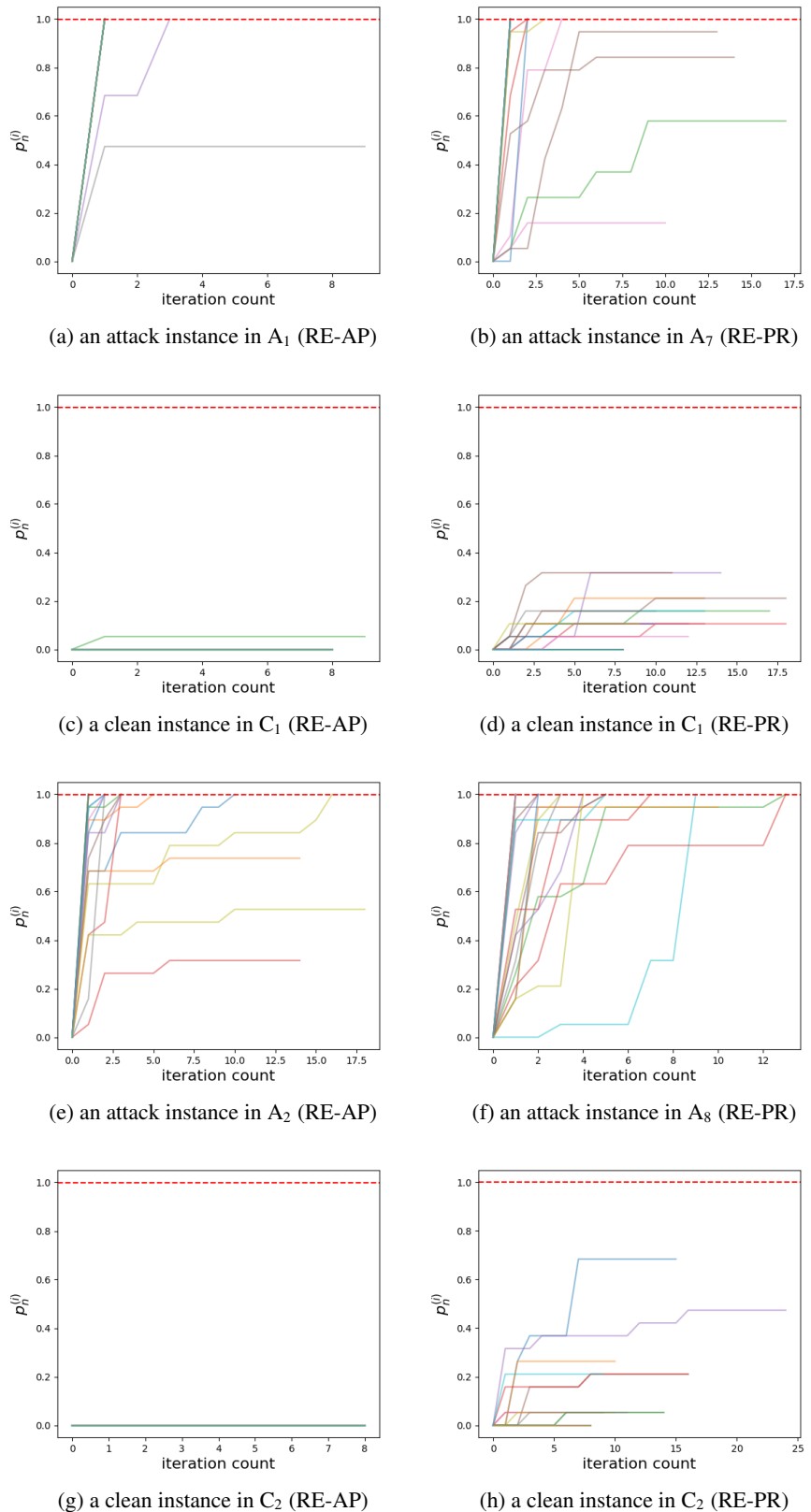

Figure 11: Example growing curves of $p_n^{(i)}$ (with patience $\tau = 8$). In each figure, there are 20 curves, each corresponding to a clean sample used for detection. ET is the average final $p_n^{(i)}$ over these 20 samples.

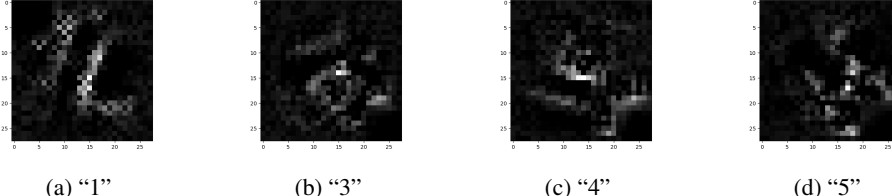

(a) "1"    (b) "3"    (c) "4"    (d) "5"

Figure 12: Images synthesized using a simpler version of the model inversion method used by Chen et al. (2019).

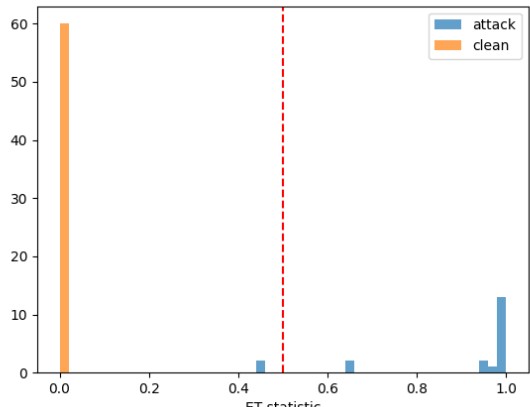

Figure 13: Histogram of ET statistics for classifiers in $A_6$ and $C_6$, when the images for backdoor pattern reverse-engineering are synthesized.

*Can our ET framework be generalized to involve backdoor pattern reverse-engineering using synthesized images?*

Here, we consider the 20 two-class domains generated from MNIST due to this domain's simplicity. We apply RE-AP to the 20 classifiers being attacked in $A_6$ and the 20 clean classifiers in $C_6$, respectively. The clean images used for detection are generated using a simpler version of the model inversion method used by Chen et al. (2019). To synthesize an image for detection, we first initialize an all-zero image added with some small random positive noise. Then, we maximize (over the image values) the posterior of the designated class of the image using the classifier to be inspected, until the posterior is greater than 0.9. Examples for our synthesized images are shown in Fig. 12. Note that without the "auxiliary constraints" on images during their generation process (Chen et al. (2019)), the generated images are usually atypical to their designated classes.

In Fig. 13, we show the ET statistic for the 80 classes associated with the 40 binary classifiers (20 classifiers being attacked and 20 clean classifiers). Among these 80 classes, there are 60 backdoor target classes and 20 non-target classes (please see the settings in Sec. 4.1). Despite one backdoor target class evading our detection and one ET for a backdoor target class lying close to the threshold $\frac{1}{2}$, the separation of ET for backdoor target classes and non-target classes is even better than that in the bottom left figure in Fig. 2 (where backdoor pattern reverse-engineering is performed on typical MNIST images). The possible reasons maybe:

- For non-attack cases, two independently synthesized samples predicted to the same class will likely be both atypical to their predicted class. They will likely share fewer features associated with their predicted class than two independent typical samples from this class. They can also be located anywhere in the input space. Thus, they will be more likely to be "mutually not transferable".

- The synthesized samples are also far from the data manifold of classes other than the class they are predicted to. So the minimum perturbation size required to induce them to be

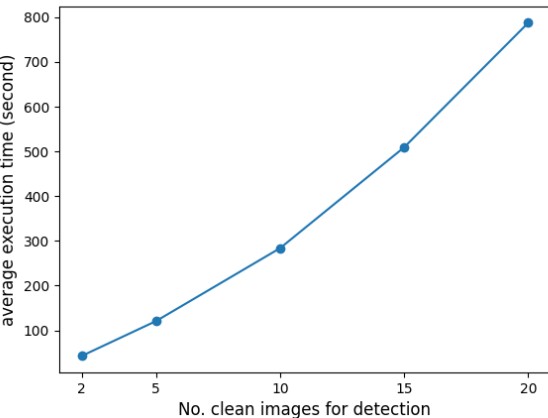

Figure 14: Execution time versus the number of samples for detection.

misclassified will have a large norm – possibly larger than the norm of the backdoor pattern when there is an attack. If this is the case, the ET statistic will be exactly 1 based on Thm. 3.3.

Future works along this research line include further investigation of the phenomenon we observed, and also improvement to the model inversion techniques, such that more complicated domains can be handled.

## J    COMPUTATIONAL COMPLEXITY

Here, we discuss the computational complexity of Alg. 1. More generally, we consider Alg. 1 for multi-class scenarios. Let $K$ be the number of classes, $N$ be the number of samples per class for detection, and $T$ be the maximum number of forward/backward propagations for backdoor pattern reverse-engineering (i.e. "solving problem (3)" in line 8 of Alg. 1). For each of the $K$ classes, we compute the $p_n^{(i)}$ quantity (line 12 of Alg. 1) for each of the $KN$ samples used for detection. To compute each $p_n^{(i)}$, the transferable set (line 9 of Alg. 1) is updated *at most* $(KN-1) \times \tau$ times; and each updating involves *at most* $T$ forward/backward propagations. Thus, the theoretical upper bound of the number of forward/backward propagations for Alg. 1 is of the order $\mathcal{O}(K^3 N^2 T \tau)$ where $\tau$ is the patience parameter.

However, the actual complexity of our method in practice is much lower than the theoretical bound. First, the purpose of using a sufficiently large number of samples for detection is to reduce the variance of the estimated ET (see Apdx. G for more discussion and empirical results). Thus, for sufficiently large $K$, we can set $N = 1$ and use only $K$ samples for detection; or even use fewer samples randomly selected from these $K$ samples.

Second, the actual number of iterations for updating the transferable set is much smaller than $(KN-1) \times \tau$ in practice. In Apdx. H, we discussed the influence of the choice of $\tau$ on the estimated ET statistic and provided empirical results. For attack cases, $p_n^{(i)}$ reaches 1 (and thus terminates the updating of the transferable set) very quickly (even in one or two iterations) for most samples (see (a)(b)(e)(f) of Fig. 11). For non-attack cases, convergence is also reached quickly for most samples (see (c)(d)(g)(h) of Fig. 11). In these examples, 20 images are used for estimating the ET statistic with patience $\tau = 8$. Thus the theoretical maximum number of iterations for the transferable set updating for an image is $(20-1) \times 8 = 152$, which is several times larger than the *actual* maximum number of iterations ($<25$).

In Fig. 14, we show the curve of the execution time growing with the number of samples used for detection. Specifically, we apply our detector to the clean binary classifiers in $C_1$ and record the average execution time, with the number of images for detection varying in [2, 5, 10, 15, 20]. Execution time is measured on a dual card RTX2080-Ti (11GB) GPU. Comparing with Fig. 10

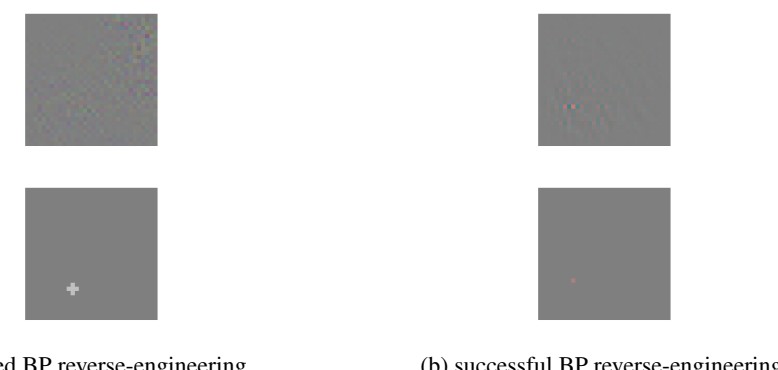

(a) failed BP reverse-engineering     (b) successful BP reverse-engineering

Figure 15: Example of (a) a failed BP reverse-engineering; and (b) a successful BP reverse-engineering. For both examples, the estimated BP is on the top, while the true BP used by the attacker is at the bottom.

where we show the effectiveness of our detector with only a few samples for detection, the actual time required for our detector to achieve good performance is only a few minutes. Moreover, for attack cases, the total number of iterations (for all samples used for detection) required for ET estimation is generally much smaller than for clean cases (as shown in Fig. 11). Thus, the actual execution time when there is an attack should be much smaller than the time shown in Fig. 14.

# K  Influence of Backdoor Pattern Reverse-Engineering on Detection Performance

Like most existing REDs, our method cannot achieve 100% detection accuracy in practice. As shown in both Tab. 1 and Fig. 2, out method, though achieving generally good detection accuracy, suffers from a few false negatives and false positives.

One main reason for the false negatives is that the existing BP reverse-engineering techniques used in our detection framework cannot not always recover the key features of the true BP used by the attacker[6]. In such cases, the "non-transfer probability" for a backdoor target class will be large, as if there were no attacks; and ET will likely be less than $\frac{1}{2}$. In Fig. 15a, we show an example of such failure of BP reverse-engineering. We consider a classifier being attacked associated with a two-class domain generated from CIFAR-100 that evaded our detection (from a blue bar less than $\frac{1}{2}$ in the second figure of the left row in Fig. 2). For a random sample, we observed that the estimated BP is visually uncorrelated with the true BP used by the attacker. For comparison, we also show an example for a successful BP reverse-engineering in Fig. 15b, where the estimated pattern contains some key features of the true BP used by the attacker; the ET statistic for this class is larger than $\frac{1}{2}$ and the attack is successfully detected.

We notice that most false positives happen when applying RE-PR (i.e. the reverse-engineering method in Wang et al. (2019) for patch replacement BPs) to clean classifiers. As introduced in Sec. D.6, RE-PR searches for a small image patch that induces high group misclassification to a putative target class. But it is possible for some domains and some classes, there are *common* key features associated with the class that is easy to be reverse-engineered on a small spatial support/mask. Such features, if reverse-engineered on one sample, will likely also induce another sample to be misclassified. Although this hypothesis requires further validation in the future, we have shown empirically that the false detections have a low frequency, given that our experiments are performed on a large number of different two-class domains generated from six benchmark datasets.

---

[6]Possible reasons can be related to the design of the BP reverse-engineering algorithm, the attack configurations (which, e.g., cause a low attack success rate), etc.

Table 10: Detection accuracy of ET with RE-AP on the 10 five-class domains with 1 attack, 2 attacks, and no attack, compared with RED-AP (original) and RED-AP (MAD).

|                  | 1 attack | 2 attacks | clean |
| ---------------- | -------- | --------- | ----- |
| ET (RE-AP)       | 10/10    | 10/10     | 10/10 |
| RED-AP (original)| 10/10    | 0/10      | 9/10  |
| RED-AP (MAD)     | 6/10     | 2/10      | 7/10  |

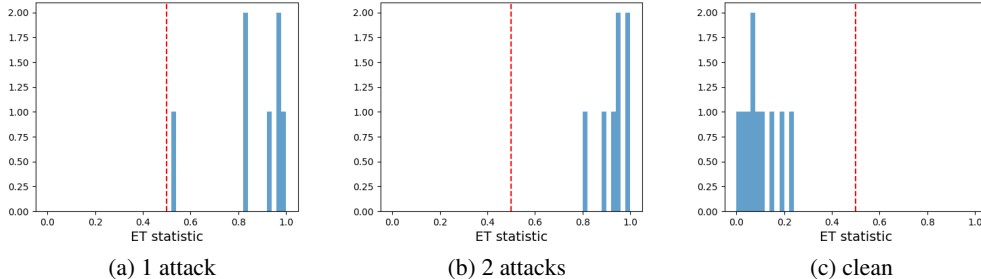

(a) 1 attack                  (b) 2 attacks                  (c) clean

Figure 16: Histogram of the maximum ET over the five classes for classifier ensembles with (a) 1 attack, (b) 2 attacks, and (c) no attack.

## L    ADDITIONAL EXPERIMENTS FOR MULTI-CLASS SCENARIOS WITH ARBITRARY NUMBER OF ATTACKS

In Sec. 4.4, we showed the performance of our ET framework against BAs for multi-class scenarios with arbitrary number of attacks. We considered the original domains of CIFAR-10, CIFAR-100, and STL-10. For each domain, we showed that the maximum ET statistic over all classes is larger than $\frac{1}{2}$ if there is an attack, and less than $\frac{1}{2}$ if there is no attack.

Here, we further investigate the capability of our ET framework on more multi-class domains. In particular, we generate 10 five-class domains from CIFAR-10, with the five classes for each domain randomly selected from the original ten classes of CIFAR-10. For each domain, we create an attack instance with one attack, an attack instance with two attacks, and a clean instance. The protocols for attack creation, classifier training, and defense configurations are the same as in Sec. 4.4. Moreover, we compare our ET (with RE-AP) with the existing RED proposed by Xiang et al. (2020) (with its original protocol including a confidence threshold 0.05 (indicating a confidence 0.95)), as well as the same RED proposed by Xiang et al. (2020) but with the anomaly detection method changed to the one based on median absolute deviation (MAD) proposed by Wang et al. (2019) (with the same threshold 2 (also indicating a confidence 0.95) used by Wang et al. (2019)). For simplicity, we name these two methods as "RED-AP (original)" and "RED-AP (MAD)" respectively. All three methods use 3 clean images per class for detection.

In Tab. 10, we show the detection accuracy of our ET compared with RED-AP (original) and RED-AP (MAD). Our ET detects all attacks with no false detections. For each of the three ensembles with ten classifiers, the histogram of the maximum ET statistic over all the five classes is shown in Fig. 16. On the other hand, RED-AP (original) achieved good accuracy to detect classifiers with 1 attack, with very low false detection rate, but it fails to detect any classifiers with 2 attacks. This is because the anomaly detection setting of RED-AP (original) relies on the 1-attack assumption to estimate a null distribution for non-backdoor class pairs. For RED-AP (MAD), there is also a clear gap in performance compared with our ET. This is because the median of the five statistics is largely biased by statistics associated with backdoor target classes.

