# OpenReview forum: "Post-Training Detection of Backdoor Attacks for Two-Class and Multi-Attack Scenarios"
_ICLR.cc/2022/Conference — ICLR 2022 Poster_

### Official Review · Reviewer_FJ6Z · 2021-11-01

**Correctness:** 4
**Technical Novelty And Significance:** 4
**Empirical Novelty And Significance:** 4
**Recommendation:** 8
**Confidence:** 3

**Main Review:**

Pros:
1. The math derivation of ET and detection threshold is convincing. With a constant threshold, ET can be used to detect different backdoor attacks in different domains.
2. The proposed BA detection algorithm using empirical estimate of ET can be applied to classifiers with arbitrary number of classes including two-class cases.
3. Authors conducted comprehensive experiments on six benchmark databases using different backdoor patterns to demonstrate the performance of proposed algorithm.

Cons:

The technical part of proposed BA detection algorithm is interesting and convincing for me. I am curious about the patience parameter to determine convergence. Does the choice of 4 hold true for ensembles A1 to A10, C1 to C6 in the experiment? Does the number of clean images used during detection affect the detection accuracy? Is the a minimum number of clean images required for an effective detection?

**Summary Of The Paper:**

This paper proposes a novel post-training backdoor attack detection algorithm using reverse-engineering defense. The authors design a new statistic called expected transferability (ET) for detection. The advantage of ET is that the detection threshold is a constant (1/2) and is independent of classifier domain and attack algorithms, so the proposed detection algorithm can be applied to a wide range of backdoor attacks on different databases.

**Summary Of The Review:**

I will vote for accepting this paper. The proposed ET statistic is novel and technically convincing. Extensive experiments have been conducted to evaluate proposed BA detection algorithm.

---

> ### Author Response · Authors · 2021-11-16
> **Response to Reviewer FJ6Z**
>
> __Thank you for your effort reviewing our paper, your supportive comments, and your recognition of our contributions.__
>
> Indeed, we are solving a very difficult practical problem – detecting backdoor attacks without access to the classifier’s training set or to any clean (or backdoor attacked) classifiers for reference, for two-class and multi-attack scenarios (as well as for multi-class, multi-attack scenarios). Our proposed ET statistic, as you said, works effectively with a constant (theoretically grounded) threshold, which allows our method to be applied to "a wide range of backdoor attacks on different databases".
>
> __We also thank you for your questions/suggestions which are constructive to our paper revision. Please find below our answers to each of your questions and the summary of corresponding paper revisions:__
>
> > __Q1: About the patience parameter.__
>
> __A1:__ In general, our choice of the patience parameter (set to 4) is not critical to the detection performance in our experiments. To better answer your question, we added a new Apdx. H (page 28, revised paper) with some empirical results and related discussion. In summary, for most attack instances and for most samples used for detection, $p_n$ (defined on line 12 of Alg. 1) grows to 1 in very few iterations, such that even for a small patience parameter, the estimated ET will likely be greater than $\frac{1}{2}$. On the other hand, for most clean instances and for most samples used for detection, $p_n$ stays at 0 or quickly grows to some small value close to 0, showing a tendency toward quick convergence. The estimated ET for these clean-cases will not change much (likely staying below $\frac{1}{2}$) even with larger choice of the patience parameter.
>
> > __Q2: About the number of images used for detection.__
>
> __A2:__ The number of clean images used for detection _does_ affect the detection accuracy. This is because the nature of our ET statistic is an expectation. With few images, estimation of ET suffers from large variance; thus, it is possible (but not often the case) that  ET will be estimated larger than $\frac{1}{2}$ when no attack is present (or will be estimated less than $\frac{1}{2}$ for the attack case). We added a new Apdx. G (page 27, revised paper) to investigate this phenomenon empirically. We show on ensemble A1 of attack instances and ensemble C1 of clean instances that, with only 2 clean images per class (4 images in total), our method achieves nearly 80% accuracy to detect backdoor attacks with less than 10% false detection rate.
>
> __We hope we have answered all your questions. Please let us know if you have any follow-up comments.__

---

### Official Review · Reviewer_xYbV · 2021-11-02

**Correctness:** 3
**Technical Novelty And Significance:** 2
**Empirical Novelty And Significance:** 2
**Recommendation:** 6
**Confidence:** 2

**Main Review:**

What I liked:
- The paper is well organized.
- The use of a class-specific metric to figure out if a classifier is prone to backdoors, given existing works focus on inter-class metrics, is a novel idea. (Although, I have not been proactive in following work on these lines.)
- I liked the idea of theoretically motivating the value for a metric and then empirically demonstrating its usefulness.
- The authors conduct several empirical studies.

Weakness/Clarification:
- Some parts of the theoretical analysis to determine the value of ET seem to be hand-wavy. First, while it is intuitively understandable why Property 3.1 might hold in the context of non-backdoor-attack scenarios, it is not clear whether it will always hold. Second, there is no good definition of how small $\epsilon$ has to be for the reasoning to hold.
- The authors argue that existing approaches cannot work for 2-class classification problems (as their sample complexity is $O(K^2)$, $O(K)$ where $K$ is the number of classes). While I get that a method that needs fewer examples is better, how many samples are too few or too many? The author's approach uses 20 samples (why 20?), is that constant or is it ~$O(K^3)/O(K^4)$ for the 2-class classification task. Is there a theoretical bound here?
- In the empirical section, there are several cases where a backdoor-poisoned classifier is determined to be benign as per the ET metric (16/20 for $A_3$ in Table 1) or vice-versa (16/20 for $C_4$). Is there any particular reason as to why this happens? Also, is there a confidence value associated with the metric (for example, would values far away from 1/2 strongly indicate that it is benign or backdoor-poisoned, or does no such correlation hold)? Looking at Fig 2, Col 3, I notice a few oranges close to 1, which signals that there are no guarantees of any kind.
- How important is it to have a constant threshold (for ET-based methods) vs. estimate the boundary with a few samples (for other baselines)? Also, from Fig 2, it seems L2, when an optimal threshold value is used, might have better separability than E2. Yet, the ROC curve tells ET is much better than L2; what was the threshold used for L2 and how was is calculated?
- Why do the authors simply show the max ET values in Table 2? I would be more interested to see aggregate statistics (similar to Table 1). The latter would give a better idea of how the ET-based approach performs on average (as opposed to in the best case, that too if values furthest from 1/2 imply best).

**Summary Of The Paper:**

The paper proposes a class-specific metric called Expected Transferability (ET) that can determine if a classifier is targeted by a backdoor attack (or not). The threat model assumes no access to the model's training data. By the virtue of being class-specific, the metric allows one to examine classifiers regardless of the number of output classes. The author reason theoretical what values of ET indicates the presence (or absence) of backdoor attacks. Then they empirically demonstrate how this value can be estimated and leveraged.


**Summary Of The Review:**

See above.

*[Update]* Having read the rebuttal (and looking at the additions to the revised paper), I would like to thank the authors for showcasing an honest effort to answer/address many of the comments/clarifications I initially had. I have updated my score to reflect the same.

---

> ### Author Response · Authors · 2021-11-16
> **Response to Reviewer xYbV (Part 3)**
>
> >__Q4: About the importance of having a constant threshold for ET (and other statistic types).__
>
> __A4:__ Having a constant threshold for backdoor detection irrespective of the domain, the DNN architecture, the attack configurations, and the backdoor pattern reverse-engineering algorithm is the _most important contribution_ of our detection framework using ET.
>
> As we have discussed in our previous letter, in our (very practical) detection scenario, there are no backdoor-free classifiers trained on the same domain available to the defender for reference. For classification domains with two classes, where both classes can possibly be a backdoor target-class, the defender also cannot estimate a null distribution for statistics associated with non-backdoor classes using the classifier to be inspected (as we explained in the first letter).
>
> Without such reference/supervision, a constant threshold on the detection statistic becomes very important. For example, although for each group of two-class domains generated from the same benchmark dataset, the L2 norm statistic (in Fig. 2) shows similarly good separation as our ET statistic, the defender still has no idea whether an L2 norm of 1 indicates an attack or not for a new domain s/he has never considered before. That is, the defender has no sound way to set the detection threshold using the L2 norm statistic.
>
> Even if we assume that the defender knows (or has successfully estimated) the distribution of the L2 norm for both attack cases and clean cases for some domains, there is no guarantee for the defender to accurately detect backdoor attacks for a new domain (using the same threshold). For example, for the two-class domains generated from F-MNIST (the 5th subfigure in the 2nd column of Fig. 2), an L2 norm of 1 is associated with a typical _non-backdoor target class_. However, for two-class domains generated from MNIST, an L2 norm of 1 is associated with a typical _backdoor target class_ – this is the case even though images from F-MNIST and MNIST are of similar styles (both gray-scale and with same size).
>
> Finally, we didn’t specify any particular detection thresholds for the ROC curves in Fig. 3. The ROC curve for each statistic is plotted using all the histograms in Fig. 2 associated with that statistic. In the revised paper, we changed the caption of Fig. 3 (page 9) to make this point clear. Please note that our ET statistic has the largest area under the curve because, unlike other statistics, there is a constant threshold for ET that is generally independent of the domain, the DNN architecture, the attack configurations, and the backdoor pattern reverse-engineering algorithm being used.
>
> > __Q5: About the experiments for multi-class, multi-attack scenarios and results in Tab. 2__
>
> __A5:__ The detection accuracy shown in Tab. 1 (i.e. correct/total) is not simple aggregation of the results from repeated experiments on the same domain. For example, for A1, each of the 45 classifiers is trained for a _different_ two-class domain generated from CIFAR-10. We do so to evaluate our method for a sufficiently large number of _different_ classification domains. And the purpose is to show that our constant ET threshold is generally domain-invariant.
>
> However, in Sec. 4.4, we considered the original domains of CIFAR-10, CIFAR-100, and STL-10 to demonstrate the ability of our ET framework for the more general multi-class, multi-attack scenarios. So we didn’t repeat the same procedure on these domains, but rather showed the maximum ET statistic that directly indicates "a detection" or "no detection".
>
> Still, we now provide aggregate results you expect in the new Apdx. L (page 32, revised paper). We created 10 different five-class domains from CIFAR-10. Similar experiments in Sec. 4.4 are conducted on these five-class domains. We also compared our method with some existing REDs to show its superiority in multi-class, multi-attack scenarios when the number of backdoor target classes is not much less than the total number of classes.
>
> __We hope you are satisfied with our responses above. Please let us know if you have any follow-up comments.__

---

> ### Author Response · Authors · 2021-11-16
> **Response to Reviewer xYbV (Part 2)**
>
> > __Q2: About the number of clean samples used for detection and computational complexity__
>
> __A2:__ First, the reason why existing REDs cannot be applied to two-class domains is explained in our previous letter – it is due to the lack of statistics for null distribution estimation, but _not_ the complexity of these methods.
>
> Second, we used 20 samples per class for detection simply because most existing REDs choose a similar number of samples for detection too. However, we agree to you that a discussion about the number of samples used for detection will be an improvement to our paper. Thus, we added a new Apdx. G (page 27, revised paper) containing related discussion with empirical results. We show on two-class domains generated from CIFAR-10 that with only 2 clean images per class (4 images in total), our method achieves nearly 80% accuracy to detect backdoor attacks with less than 10% false detection rate.
>
> Third, we also added a new Apdx. J (page 31, revised paper) to discuss the computational complexity of our method. For $K$ classes, $N$ clean samples per class for detection, at most $T$ forward/backward propagations (for backdoor pattern reverse-engineering), and patience parameter $\tau$, the upper bound of our computational complexity is of order ${\mathcal O}(K^3 N^2 T \tau)$. However, the _actual_ execution time of our method in practice will be _far less_ than the theoretical upper bound, since: a) the required number of clean samples for detection need _not_ grow with the number of classes $K$; b) the number of iterations required for the transferable set to remain unchanged for $\tau$ iterations is usually small. Please find more details in Apdx. J. Moreover, our method is proposed mainly for the cases where there are not many classes in the classification domain but possibly multiple attacks; thus, the scalability of our method to the number of classes is not a big concern. Finally, our method is deployed off-line and only needs to be applied once. A few minutes (as shown by our empirical results in Fig. 14 on page 31) on a regular PC should be an acceptable execution time.
>
> > __Q3: About the false positives and false negatives__
>
> __A3:__ In general, our method has achieved high detection accuracy with few false detections. In the new Apdx. G (page 27, revised paper), we also show that similarly good performance can be achieved with even fewer clean images for detection. As in our response to your comments regarding theoretical analysis, deriving a confidence bound on ET is extremely hard, due to a lot of factors including the choice of the backdoor pattern reverse-engineering algorithm and its associated configurations.
>
> However, thanks to your comment, we added a discussion in the new Apdx. K (page 32, revised paper) to explain the occurrence of those false positives and false negatives. We believe that the false negatives are due to the occasional failure of the reverse-engineering method to recover the true backdoor pattern used by the attacker. In this case, the pattern estimated for one sample, since it is not correlated with the backdoor pattern, cannot induce other samples from the same class to be misclassified. Thus, the ET will likely be less than $\frac{1}{2}$, causing a false negative.
>
> On the other hand, the reason for false positives may be due to the existence of some common key features that are easy to be reverse-engineered on a small spatial support associated with some classes in the domain. Although this hypothesis requires further validation in the future, we have some belief in this explanation since these false positives are domain-dependent. Note that in each subfigure of Fig. 2, the histogram is plotted for ET statistics obtained for a number of different two-class domains, but not ET statistics obtained from repeated experiments on a single two-class domain. For the domain associated with the "oranges close to 1" as you mentioned, repeated experiments using another set of clean samples for detection still yield ET statistics close to 1.

---

> ### Author Response · Authors · 2021-11-16
> **Response to Reviewer xYbV (Part 1)**
>
> __Please find below our responses to your comments/questions and the summary of the revisions we made.__
>
> > __Q1: About the theoretical analysis__
>
> __A1:__ As you said, the purpose of the theoretical analysis in the main paper is to motivate the design of our detection framework using the ET statistic. We agree that deriving a "provable/certified" bound on detection confidence or error rate for using the constant ET threshold $\frac{1}{2}$ will be an important future work that will add huge values to our current work. But given the difficulty of the problem we are solving, such derivation is even tougher due to: 1) the complicated data distribution, 2) the complicated DNN architecture, 3) the wide range of possible attack configurations, and 4) the randomness caused by the backdoor pattern reverse engineering algorithm being used. Thus, we presented Property 3.1 based on empirical results from many existing works to support the use of the constant ET threshold. More importantly, whether this property is guaranteed to hold in general, we have shown strong empirical support of its holding in practice by our experimental verification in Sec. 4.3 (page 8), and also by the effectiveness of our method in our experiments in Sec. 4.1 that involves 145 different two-class domains with a broad range of attack configurations.
>
> For similar reasons, we also didn’t include quantitative analysis regarding the solution gap $\epsilon$ but only stated the fact that it is typically small for existing backdoor pattern reverse-engineering algorithms. We discussed the asymptotic behavior of ET (through the $P_{\rm MT}$ term) when $\epsilon$ is close to 0 (by Thm 3.2 on page 5) though. Again, the confidence bound for ET being less than $\frac{1}{2}$ for a given $\epsilon$ may need some strong assumptions on the data distribution and DNN architecture – derivation of such bound for general practical scenarios is extremely hard, so we defer it to our future work.
>
> Nevertheless, we did include a section in Apdx. B (page 13 of the first submission and the revision), where we show for a simpler yet still relatively general problem that $\frac{1}{2}$ is a strict upper bound of ET when there is no backdoor attack. Some analysis in this section can possibly inspire future developments (by gradually removing some assumptions) of the theoretical basis of our detection framework.

---

> ### Author Response · Authors · 2021-11-16
> **A letter to Reviewer xYbV**
>
> Thank you for your time reviewing our paper and recognizing the novelty of our method. We also thank you for offering us a chance to make clarifications to your concerns about our work. Before answering your questions, we want to emphasize the difficulty of the problem we are solving, and our contributions compared with the existing works, in this first letter. __Some of the contents here are also useful in helping to answer your questions (which we do explicitly in the second letter (below)).__
>
> As we have mentioned in the 2nd paragraph of Sec. 1, detecting backdoor attacks without access to the training set is an important problem. In scenarios such as proprietary classifier systems (e.g., where the government purchases a classifier from a company but does not possess data rights), legacy systems (where the training set has long been forgotten), and e.g. classifier apps on a cell phone, the data set used to train the classifier will not be available to the defender. This problem is also challenging, since without access to the possibly poisoned training set, the defender has no idea what the backdoor pattern used by the attacker (if there is an attack) will look like (i.e. __S1__ on page 3).
>
> In the same scenarios mentioned above, the defender also has no access to any backdoor-free classifiers trained for the same domain (e.g. for reference during detection, i.e. __S2__ on page 3). Usually, such backdoor-free classifiers do not even exist since the training set has already being poisoned. Even if they do exist and are available to the user, s/he will not likely use them for detection since there is no need for a backdoor defense – these known-to-be clean classifiers can be directly used for class decision-making.
>
> To address such a challenging problem with constraints S1 and S2, a large family of __reverse-engineering defenses (REDs)__ have been proposed (e.g. the six papers cited in the 2nd paragraph on page 3). As introduced in Sec. 2, REDs trial reverse-engineer the backdoor pattern to address S1; and address S2 by performing anomaly detection using statistics derived from the estimated backdoor patterns. For example, for a classifier on CIFAR-10 with 10 classes, an existing RED first estimates a pattern for each of the 90 ordered class pairs. If there is no attack, the L2 norm of all these estimated patterns can be approximated by (e.g.) a normal distribution centered about 10 with standard deviation (e.g.) =2. Now consider there is a backdoor attack with source class $s$, target class $t$, and a backdoor pattern with L2 norm (e.g.) =1 (the norm of backdoor patterns are usually small for imperceptibility). If the RED successfully reverse-engineers the backdoor pattern for class pair $(s, t)$, the L2 norm of the estimated patterns for the other 89 class pairs will still form a normal distribution near 10, while there is an outlier near 1 corresponding to class pair $(s, t)$. Then, the existence of such an anomaly triggers the alarm.
>
> Existing REDs rely on the assumption that if there is an attack, the number of backdoor target classes is much less than the total number of classes. Thus, there will be a sufficient number of statistics (e.g. the L2 norm in the example above) to reliably estimating a null distribution (for assessing anomalies). However, for two-class domains, existing REDs are not applicable due to the following reasoning. For the two statistics obtained for the only two classes respectively, the atypicality of one statistic need to be assessed by a null distribution estimated using the other statistic, and vise versa. But one sample (i.e. statistic) is certainly inadequate for estimating a null distribution – one can get an (albeit unreliable) estimate of the mean using one sample. One cannot estimate the variance at all using one sample. So existing REDs are not applicable to the two-class case. And while they are applicable when there are more than two classes, estimation of the null distribution will be inaccurate when there are only few classes in the problem – this is also reflected by our empirical results in the new Apdx. L (page 32, revised paper) added in response to your comments on our multi-class experiments (which will be detailed in our next letter).
>
> Thus, by focusing on the two-class, multi-attack scenarios (i.e. __S3__ on page 3), with also S1 and S2 (considered by existing REDs), we are solving a very difficult and important problem. Moreover, to the best of our knowledge, we are the first to address this challenging scenario. Our work will potentially inspire more studies along this research line.

---

### Official Review · Reviewer_BBKY · 2021-11-02

**Correctness:** 2
**Technical Novelty And Significance:** 2
**Empirical Novelty And Significance:** 2
**Recommendation:** 5
**Confidence:** 4

**Main Review:**

* Related work and novelty claims

It appears to me that the paper does not include discussions on recent work. Its
key selling point is that the proposed method works under three conditions: no
access to training data, reference model, and binary classifier.

Many existing work also work under these constraints, such as ABS proposed by
Liu et al. in CCS 2019. More broadly, the IARPA TrojAI competition include not
only these constraints but more, such as constrained time to finish scanning one
model etc. It involves using a private dataset for training backdoor models
(a.k.a., detectors cannot leverage training data or train their reference
models), and also many tasks are binary classifiers, e.g., sentiment analysis.
Many submitted solutions are based on reverse-engineering. Also, as a
requirement of this competition, all solutions are open sourced.

Thus, to me, I think many claims in this paper should be altered. Moreover, I
think the paper should compare with some of these solutions. I would suggest
using the TrojAI dataset (also open sourced on NIST web page) as their benchmark
and compare with the leaderboard results (also open to access on NIST web page)
so that we can better understand the effectiveness of proposed solution.

* Design of ET and analysis

Despite the formal representation of the method and discussion (which is
appreciated), the design of this method is based on empirical observations on
the transferability of input samples. As such, there is no guarantee that the
proposed method will work and the threshold value 1/2 is general.In fact, as
shown in Figure 2, the proposed method also suffers some FP/FN, namely, some ETs
for benign classes are larger 0.5 and ET of backdoor target classes can be
smaller than 0.5.

To me, it is important to understand the applicable scenarios of the proposed
method. Namely, why these cases fail? I personally suspect it is because of the
transferability assumption. Is it? I would strongly suggest putting such
assumptions upfront so that we can understand the pros and cons of this method.

Another concern I have is that, many existing method does require tuning. But if
we can tune them to get better results, what is the advantage of the proposed
method? More importantly, are there backup plans when the method fails? Or can
we know that when it will work or not? Otherwise, I do not see the fundamental
benefits of using this method.

* Evaluation for multi-class, multi-attack scenario is not sufficient.

In Section 4.4, this paper provides the evaluation results on multi-class,
multi-attack scenarios. However, it only trains one model for each attack
settings and a single clean model for each dataset. There is no baseline method,
and I am not sure if the proposed method can truly generalize to other datasets
and models.

* Countermeasures and ethical statement

There is no countermeasures discussed and no ethical statement.

* CIFAR-10

Why the maximum ET for CIFAR-100 is much larger than that of other datasets? As
shown in Table. 2, an interesting phenomenon is that the maximum ET statistic
for CIFAE-100 is much larger than that of other datasets (i.e., CIFAR-10,
STL-10). It would be better if this paper could provide some discussion about
this phenomenon.

* Code and data

Will the code and data be released?


**Summary Of The Paper:**

This paper proposes a new engineered reverse-engine based backdoor model
detection method. It has stronger assumptions including: 1) no access to
training data; 2) no access to benign reference model; 3) 2 or multi-class
models. To detect backdoors in such scenarios, it leverages a new statistical
metric known as the expected transferability, which can derive a fixed threshold
for backdoored models detection. Results on six datasets show that the proposed
method is effective for detecting backdoored models.

**Summary Of The Review:**

My main concern is the it lacks discussion and comparison with works that works
under the same setting. Its analysis is not sufficient to understand the
applicable scenarios and its key benefit.

---

> ### Author Response · Authors · 2021-11-16
> **Response to Reviewer BBKY (Part 4)**
>
> > __Q3: Evaluation for multi-class, multi-attack scenario__
>
> __A3:__ The main purpose of our experiments in Sec. 4.4 is to show the generalization capability of our method to multi-class and multi-attack scenarios. The generalization capabilities of our method to other domains and model architectures are demonstrated by our main experiments in Sec. 4.1&4.2 – our method achieves good performance on 145 different two-class domains generated from six benchmark datasets and four different model architectures.
>
> We didn’t show aggregated results in Sec. 4.4 like in Tab. 1 because what we considered was the original domains of CIFAR-10, CIFAR-100, and STL-10. To address your comment, we added a new Apdx. L (page 32, revised paper), where we tested our method on 10 five-class domains randomly generated from CIFAR-10. We not only considered 1-attack case and 2-attack case for each domain, but also compared our method with existing REDs with two different anomaly detection techniques. We showed that our method detected all attacks without false detection, and also discussed why existing REDs cannot achieve the detection accuracy as strong as ours.
>
> > __Q4: Countermeasures and ethical statement__
>
> __A4:__ We indicated that we had read the code of ethics prior to submission. In the final version of our paper, we will write:
>
> "This research is supported by XXX. The main purpose of this research is to understand the behavior of deep learning systems facing malicious activities, and enhance their safety level by unsupervised means. Code will be open-source released for research purpose only."
>
> We would also be appreciative if you can provide an example of countermeasures and ethical statement that meets the standard set of the security society.
>
> > __Q5: Results in Tab. 2 regarding CIFAR-100__
>
> __A5:__ The maximum ET for CIFAR-100 is much larger than the maximum ET for CIAR-10 and STL-10 may due to the same reason that caused the false positives in Fig. 2 – the key features associated with some classes are easy to be reverse-engineered on a small spatial support. Thus, these features, if reverse-engineered on one sample, can easily induce another non-target class sample to be misclassified to the target class. Note that CIFAR-100 has 100 classes, while the other two datasets both have 10 classes. The chance for the existence of such a class for CIFAR-100 is thus larger than for the other two datasets.
>
> > __Q6: Code and data__
>
> __A6:__ The datasets we used are all public. The code will be open-source released right after the reviewing process.
>
> __Thank you again for your time and thoughtful comments. Please let us know if you have any follow-up comments.__

---

> ### Author Response · Authors · 2021-11-16
> **Response to Reviewer BBKY (Part 3)**
>
> > __Q2: Design of ET and analysis__
>
> __A2:__ Whether Property 3.1 (regarding mutual non-transfer probability between two random samples) is guaranteed to hold in general, we have shown strong empirical support of this property holding in practice (even if not in all possible instances). Please note that our detector achieves generally good detection accuracy for a wide range of different domains generated from six benchmark datasets of different styles. This property is also validated by many existing works and verified experimentally in Sec. 4.3 in our paper.
>
> We agree that deriving a "certified/provable" bound on the error rate for using $\frac{1}{2}$ as the detection threshold for ET would be a great contribution. But this is extremely hard to obtain due to the complicated data distribution and DNN architecture, the wide range of possible attack configurations, and the randomness caused by the backdoor pattern reverse engineering algorithm being used in practice. Still, we showed for a simplified yet relatively general problem (with some assumptions) that $\frac{1}{2}$ is a strict upper bound of ET when there is no backdoor attack (Apdx. B, page 13 of the initial submission). In our future work, we will try to provide a stronger certification to this method by removing assumptions from our analysis on the simplified problem.
>
> To address your comments about the FP/FNs in our results, we added a new Apdx. K (page 32 of revised paper) to discuss the possible reasons. In summary, the false negatives are mainly due to the occasional failure of the reverse-engineering method to recover the true backdoor pattern used by the attacker – this has been validated by some empirical results in Apdx. K. The false positives, on the other hand, may due to the fact that key features associated with a class may be easy to be reverse-engineered on a small spatial support/mask. Such features, if reverse-engineered on one sample, will likely also induce another sample to be misclassified.
>
> As for tuning-based methods, they typically require many more input samples than most detection-based methods. For example, the "unlearning" method mentioned in the famous Neural Cleanse paper uses 500 samples per class; while only a few or tens of clean samples per class are required by our method (Fig. 10, page 27, revised paper) and many other REDs. Also, as mentioned in the paragraph of "Backdoor Defense" on page 2 (initial submission), tuning-based methods usually suffer from degradation in the classifier’s accuracy on clean samples during testing (e.g. Fine-Pruning (Liu et al), which is a combination of fine-tuning and neuron pruning). Thus, in practice, one should not apply a tuning-based method to a classifier before assessing whether it is backdoor-attacked. However, we agree with the reviewer about the importance of tuning-based methods. Thus, in the last two lines of the first paragraph on page 3 (revised paper), we mention that tuning the classifier deemed to be attacked leveraging the estimated backdoor pattern can help the classifier to "unlearn" the backdoor mapping.
>
> Finally, any method may fail – even tuning may not fully remove the backdoor mapping. Typically, a defender will not know if the attack on a classifier has been eliminated/corrected until a test-time misclassification due to the embedded backdoor pattern occurs. So the best choice of the defender is to deploy multiple defenses sequentially or in parallel. For example, our method detects backdoor attacks off-line. It can be combined with a defense during operation of the classifier, e.g. [2], to detect if a test sample contains a backdoor or not. As for the scenarios where our method may not work well, based on our hypothesis for the FP/FNs, there may be two. First, for some domains, there may be classes semantically close to each other, such that backdoor pattern reverse-engineering may find other features uncorrelated with the true backdoor pattern used by the attacker. Then, the ET statistic will likely be less than $\frac{1}{2}$, causing a FN. Second, for some domains, there may be classes with strong key features that are easy to be reverse-engineered. As in our previous response, the ET statistic may be larger than $\frac{1}{2}$, causing a FP. However, since our ET framework can be easily incorporated with many backdoor pattern reverse-engineering techniques, the above limitations may be resolved by future developments of these techniques.
>
> [2] Y. Gao, C. Xu, D. Wang, S. Chen, D. C. Ranasinghe, and S. Nepal. STRIP: A defence against trojan
> attacks on deep neural networks. In Proc. ACSAC, 2019a.

---

> ### Author Response · Authors · 2021-11-16
> **Response to Reviewer BBKY (Part 2)**
>
> __b)__ About the time constraint
>
> Execution time for detection is certainly an important metric for performance evaluation. But please note that the scanning time requested by the TrojAI competition is actually the inference time; while the time required for training the software (e.g. a meta classifier) can be super large. In addition, it is very possible that the software should be re-developed if the defender needs to perform backdoor detection for a new domain. However, our method requires just a few minutes (which is acceptable as an off-line detection process) to tell if a classifier for a new domain is backdoor attacked or not (Fig. 14 on page 31, revised paper) – there is no training time (or substantial data/storage) required.
>
> __c)__ About our benchmark datasets
>
> The benchmark datasets (and the attack configurations) used in our experiments are widely used by most backdoor defense papers. Since one main contribution of our work is the _domain-independency of our constant detection threshold_ on ET, we generated a large number of two-class domains from these benchmark datasets – each two-class domain has a __unique__ pair of labels. However, the binary classifiers in the TrojAI competition are trained on the same domain with fixed set of labels, thus are not suitable for demonstrating the domain-independency of the detection threshold of our ET statistic.
>
> __d)__ About ABS and other related works (REDs)
>
> ABS is certainly an influential backdoor defense (which is also based on backdoor pattern reverse-engineering) – thanks for your suggestion, we have now cited this important work in the 2nd paragraph on page 3. For each target class, ABS trial reverse-engineers a backdoor pattern on a group of clean inputs from all other classes using a novel objective function involving neuron activations hypothesized to be caused by the backdoor mapping. The estimated pattern is embedded into another group of clean inputs from classes other than the target class; and the misclassification fraction (named as "REASR") is used as the detection statistic. For multi-class scenarios, ABS can possibly be implemented with an anomaly detector like most other REDs. When there is one or few backdoor target classes, REASR for these classes will likely be abnormally large, and the attack will be detected.
>
> Existing REDs are not applicable to the two class domains we are focusing on. For two-class domains, REDs need to detect if there is an outlier among the two statistics, one for each class. To assess the atypicality of any one of the two statistics, a null distribution should be estimated using the other statistic. But estimating a null distribution using a single statistic/sample is certainly inadequate – one can get an (albeit unreliable) estimate of the mean using one sample – and no estimate of the variance of the distribution. This is the reason why we couldn’t compare our method with existing REDs (with their anomaly detectors) on the two-class domains generated from the benchmark datasets (in Tab. 1).
>
> However, on these two-class domains, we compared our ET statistic with the _statistics used by exisiting REDs_ to see if any of them can be used to effectively distinguish backdoor target classes from non-target classes without domain-specific supervision (since this constraint is required by most REDs and is the main reason for them to use an anomaly detector). As shown in Fig. 2, The existence of a common threshold allows our ET statistic to be more suitable for two-class domains and without domain-specific supervision. Please note that although the REASR statistic used by ABS for backdoor target classes is effectively higher than for non-target classes for most domains, its absolute value likely depends on the number of inputs used for backdoor pattern reverse-engineering, as indicated by [1],
>
> [1] S.-M. Moosavi-Dezfooli, A. Fawzi, and P. Frossard. Universal adversarial perturbations. In Proc.
> CVPR, 2017.

---

> ### Author Response · Authors · 2021-11-16
> **Response to Reviewer BBKY (Part 1)**
>
> __Thank you for your time reviewing our paper and your thoughtful comments. We are glad that you recognized the difficulty of our defense scenario where the defender only has minimum knowledge and capability.__
>
> As you summarized, the defender has no access to the training set, has no access to any reference/supervision from classifiers trained for the same domain, and the classification domain may have only two classes such that existing reverse-engineering defenses (REDs) are not applicable.
>
> __We also noticed that your major concern of our paper is regarding the comparison with other backdoor defenses. We hope our clarification below to your major concern (and also other comments) will change your current view of our work.__
>
> > __Q1: Related work and novelty claims__
>
> __A1: a)__ About TrojAI
>
> We have been involved in the discussion of this IARPA BAA a long time before the first round of the TrojAI competition was posted. The TrojAI competition solves a different problem than ours.
>
> TrojAI solves a __supervised__ problem. Based on the description of the dataset (https://data.nist.gov/od/id/mds2-2195), there is a training set consists of "trained AI models", where a "known percentage" of these models "have been poisoned with a known trigger". The task is to "develop software solutions for detecting which AI models have been poisoned". So, basically, the task is to build a software (e.g. a meta classifier) to distinguish classifiers "with attack" and classifiers "without attack", using a provided training set containing classifiers from both categories.
>
> However, the nature of the problem we are solving is __unsupervised__. The defender might be a cell phone user who downloads an app (with a classifier) from online and wants to know if the classifier has been backdoor attacked or not – clearly this is a common practical scenario of great importance. Different from the defender in the supervised TrojAI problem, our defender: __1) does not know the trigger, 2) does not have other classifiers (either "with" or "without" attack) for reference/supervision;__ but needs to tell if this single classifier to be inspected is backdoor attacked or not.
>
> Due to the above fundamental differences, our (unsupervised) method is not comparable with the method proposed for the supervised problem of TrojAI. However, there are a lot of methods proposed to solve the unsupervised problem as we do. These methods (named "RED" for "reverse-engineering defense" in the paper) usually contain a trigger/backdoor pattern reverse-engineering step to address the challenge that __the trigger is unknown;__ and an anomaly detection step to address the challenge of __no supervision.__ We not only discussed how REDs work in detail in Sec. 2 (please see the six REDs we cited in the 2nd paragraph on page 3 of our initial submission, and even more discussions in Apdx. D6&D7), pointed out their limitations (3rd paragraph on page 3), but also compared with some of them in our experiments in Sec. 4.2. Beyond REDs, we also discussed other defenses (the "Backdoor Defense" paragraph in Sec. 2) that allow the defender to have more capabilities (e.g. sufficient clean data or training capabilities/storage).

---

### Official Review · Reviewer_Xbfh · 2021-11-03

**Correctness:** 3
**Technical Novelty And Significance:** 3
**Empirical Novelty And Significance:** 3
**Recommendation:** 8
**Confidence:** 3

**Main Review:**

This paper proposed a novel and working solution to a tough problem of backdoor attacks detection with binary or multiple labels.

Strengths:
* A novel notion of detection statistics called 'expected transferability' (ET) is proposed to help in detecting the target labels.
* A domain/attack/network agnostic threshold (1/2) is used which is principled and has theoretical implications.
* This approach is extendable to address more challenging scenarios, with a potential computation cost.

Concerns:
* One argued contribution of the work is that it does not need 'the training set of the classifier to be inspected'. However, the approach still requires collecting an individual set of clean samples. I think it would be better to remove this claim.
* My major concern for this problem is that it seems very slow for the proposed algorithm to converge, as $ET$ is estimated by iterating each class $i$ and each of its samples $x_n^{(i)}$, until transferable set remain unchanged. Since the same procedure needs to be done for each class label, this issue might propagate with more class labels or clean samples.

Minor comments:
Page 4, definition 3.3 , the expectation $E$ should be over $X, Y \sim P_i$ instead of  $X \sim P_i$?

**Summary Of The Paper:**

This paper proposed a post-training BA detection approach for two-class classification problems, which can also be extended to scenarios with multiple attacks. The approach is derived based on a proposed notion of 'expected transferability', which works with a principled threshold that is irrespective of the domain nor the network or attack configurations. Empirical experiments verified the effectiveness of their approach.

**Summary Of The Review:**

This paper proposed a novel and working solution to a tough problem of backdoor attacks detection with binary or multiple labels. The solution is well motivated and theoretically justified. Although the convergence rate might be a potential drawback of this approach, it still makes solid contributions to the community.

---

> ### Author Response · Authors · 2021-11-16
> **Response to Reviewer Xbfh**
>
> __Thank you for your effort reviewing our paper. We are appreciative that you recognize the novelty of our method and, especially, the difficulty of the problem we are solving. We  thank you for raising some concerns that helped us to further improve our paper. Here are our responses and the summary of our corresponding paper revisions:__
>
> > __Q1: About clean samples used for detection__
>
> __A1:__ First, allowing the defender (e.g. a smart phone user who wants to check if a downloaded app with a classifier is backdoor-free) to possess an independently collected, small, and clean dataset is a relatively mild assumption commonly used by most related works. In fact, it seems to be a common practice for many (e.g.) smart phone users to collect some clean samples to validate their new app upon downloading. For example, many people using Siri for the first time tend to check if their voice can truly be recognized. Moreover, the number of clean samples required for detection is in general quite small  (e.g. 20 images per class in our experiments). In the new Apdx. G (page 27, revised paper), we demonstrate the effectiveness of our method with even fewer clean images for detection.
>
> Also, following your comment, we believe it is a good future work research direction to seek to remove the need for an independent, clean data set. In fact, in the new Apdx. I (page 28, revised paper), we borrow the idea from [1] to synthesize "clean images" for detection using model inversion techniques [2]. The preliminary results on MNIST show that our method works well using synthesized images for detection.
>
> [1] H. Chen, C. Fu, J. Zhao, and F. Koushanfar. DeepInspect: A Black-box Trojan Detection and
> Mitigation Framework for Deep Neural Networks. IJCAI, 2019.
>
> [2] M. Fredrikson, S. Jha, and T. Ristenpart. Model inversion attacks that exploit confidence information and basic countermeasures. CCS, 2015.
>
> > __Q2: About the convergence rate__
>
> __A2:__ Your summary of our ET estimation procedure is accurate. Theoretically, the complexity for ET estimation grows with the number of classes $K$, the number of images used for detection $N\times K$ ($N$ images per class), and also grows with the patience parameter $\tau$. However, in practice, we only need a sufficient number of clean images to reduce the variance of our ET estimation for each class. This number need _not_ grow with the number of classes. Again, in the new Apdx G (page 27, revised paper), we have shown the good performance of our method with very few clean images for detection. Moreover, the number of iterations required for the transferable set to remain unchanged for $\tau$ iterations is usually small (much smaller than the theoretical maximum). Especially, for many attack cases, $p_n$ (line 12 of Alg. 1) reaches 1 in very few iterations (which terminates the “while” loop in Alg. 1). Empirical growth curves for $p_n$ (for both attack cases and clean cases) are shown in Fig. 11 (page 29, revised paper).  Another point to make is that our detector needs to be applied just once for a given trained classifier (to be inspected).  And while the time required to make accurate detection inference may be significant for a domain with many classes, the classifier training already requires substantial computation (it is essentially "off-line" training) – thus, use of our detector does not fundamentally change the nature of training+certification as an "off-line" learning approach.
>
> To better address your concern, in the revised paper, we added a new Apdx. J (page 31, revised paper) to discuss the computational complexity of our method. Moreover, we showed the empirical execution time for our method applied to binary classifiers for a range of clean images per class for detection.
>
> Finally, like all existing REDs, the complexity of our method does grow with the number of classes, since all classes are inspected as a putative backdoor target class. We agree that improving time efficiency is an important topic in this research line. But our method is proposed to address the scenarios where the number of backdoor target classes are not much less than the total number of classes, e.g. domains with two classes.
>
> > __Q3: Minor comments about notations in Def 3.3__
>
> __A3:__ For $X$, $Y$ i.i.d. following $P_i$, the conditional probability will be a function of $X$. So the expectation is actually taken over $X$. But we agree to you that the notation in our initial submission may cause confusions to readers. Thus, we remove the subscript $X\sim P_i$ since it has already been stated in the front of Def 3.3.
>
> __Thank you again for your thoughtful review. Please let us know if you have any follow-up comments.__

---

### Decision · Program_Chairs · 2022-01-20

**Decision:**

Accept (Poster)

**Comment:**

The paper presents a new method for detection of backdoor attacks under strong limitations such as the lack of access to training data and the reference benign model. Its main idea is to utilize a new expected transferability statistic that can be used for detection in broad range of application domains. The effectiveness of the proposed approach is demonstrated experimentally.